# Human-Robotic Prosthesis as Collaborating Agents for Symmetrical Walking

**Ruofan Wu** *
Arizona State University

**Junmin Zhong** *
Arizona State University

**Brent Abraham Wallace**
Arizona State University

**Xiang Gao**
Arizona State University

**He Huang**
North Carolina State University

**Jennie Si** †
Arizona State University

## Abstract

This is the first attempt at considering human influence in the reinforcement learning control of a robotic lower limb prosthesis toward symmetrical walking in real world situations. We propose a collaborative multi-agent reinforcement learning (cMARL) solution framework for this highly complex and challenging human-prosthesis collaboration (HPC) problem. The design of an automatic controller of the robot within the HPC context is based on accessible physical features or measurements that are known to affect walking performance. Comparisons are made with the current state-of-the-art robot control designs, which are single-agent based, as well as existing MARL solution approaches tailored to the problem, including multi-agent deep deterministic policy gradient (MADDPG) and counterfactual multi-agent policy gradient (COMA). Results show that, when compared to these approaches, treating the human and robot as coupled agents and using an estimated human adaption in robot control design can achieve lower stage cost, peak error, and improved symmetry to ensure better human walking performance. Additionally, our approach accelerates learning of walking tasks and increases learning success rate. The proposed framework can potentially be further developed to examine how human and robotic lower limb prosthesis interact, an area that little is known about. Advancing cMARL toward real world applications such as HPC for normative walking sets a good example of how AI can positively impact on people's lives.

## 1 Introduction

The concept, design, and applications of human-robot cooperation have advanced rapidly due to new demands in AI-enabled applications fueled by powerful deep learning and reinforcement learning algorithms [1, 2, 3]. Human-robot collaboration can take on a variety of forms depending on tasks to be solved, how information is shared [1, 2], and the nature of interaction [3]. Examples of complex collaborative tasks may include picking up or carrying objects together [4, 5], cooperating on a production line, in which cases a robot can learn to imitate human demonstrations [6, 7, 8]. Other application scenarios may include intermittent robotic correction of human driving, or vice versa [9]. In essence, most of these recent studies involve interactions between a human and a robot in a way such that there is either space between the agents, or there is time for predictive counter measures to interfere. By contrast, in the HPC problem the human and robot agents are physically coupled together, and there is often little time for the human agent to react to prevent from falling or injury.

---

*Equal Contribution.

†Corresponding author: si@asu.edu

36th Conference on Neural Information Processing Systems (NeurIPS 2022).

As a result, innovative solution frameworks are needed to first address the design, analysis, and testing of complex control systems to restore locomotion for amputees. Based on these platforms, human-robot interactions can be studied, and fundamental issues such as user preference and prosthesis embodiment can be examined. The knowledge in turn, will further help design seamless automatic robot control systems. As such, challenges arising from controlling a wearable robotic lower limb prosthesis to meet the human user's needs are unique, and in some aspects, these challenges are greater than what have been studied [10, 11, 12, 13].

State-of-the-art automatic control of robotic prosthesis has been reported recently with successful human subject testing [14, 15, 16, 17]. These approaches are single agent-based reinforcement learning controls, specific tasks such as level ground and ramp walking were accomplished via designer prescribed or specified robotic joint movement profiles [14, 15]. A most recent progress in single-agent based reinforcement learning solution demonstrated that human amputee subject can perform level ground and ramp walking with the robot controller aimed at mimicking the intact joint movement [17]. Note, however, that all previous results have not directly considered human influence on the human-robot system performance in tuning the robotic prosthesis controller, a phenomena that has direct impact not only on restoring walking but also on human health [18].

Achieving normative walking is fundamentally a real time control problem that involves continuous states and continuous controls of the human-robot system. Continuous state and control problem has received great attention. Approaches based on (deep) reinforcement learning have shown promise to substantively address real world applications. Several algorithms, such as Deep Deterministic Policy Gradient (DDPG) [19], Proximal Policy Optimization (PPO) [20], Soft Actor-Critic (SAC) [21], and Twin Delayed DDPG (TD3) [22], have demonstrated success with solving complex control problems. For example, in simulated human locomotion control [23, 24], deep reinforcement learning solved over 20 independent control signals to facilitate a humanoid robot to achieve different walking tasks. Additional single agent-based continuous control has also demonstrated promise in engineering applications such as stabilization, tracking, and reconfiguring control of Apache helicopters [25, 26, 27], stabilization and control of large power grids [28, 29, 30], robotic manipulation and locomotion via MuJoCo and OpenAI Gym [31, 32], and wearable robots with human in the loop [14, 15, 16, 17].

While these works are encouraging toward solving realistic single-agent continuous control problems, it is not obvious how they can directly address the multi-agent human-robot normative walking problem as needed when we consider human influence in the robot control design. From a physical human-robot interaction (pHRI) perspective, robotic upper-limb control has undergone intense development, especially in the realms of patient rehabilitation [33] and industrial applications [34]. However, this type of pHRI problems differ fundamentally from the human-robot walking problem, as their control target usually consists of well-defined end points generated by a decoupled trajectory generation exosystem [35]. On the other hand, human-robot walking tasks are difficult to associate with an end point task goal due to tight dynamic coupling between the human and the robotic lower limb, and many factors can affect the human's performance goal. As such, even though multi-agent reinforcement learning (MARL) control is a natural candidate to address our HPC challenge, a feasible solution is yet to be developed.

## 2   Related Work and Challenges

**Single agent RL for automatic control of a robotic limb.  Simulation.** Most state-of-the-art RL control design approaches to enable continuous human-robot walking are single-agent based. Important milestones have been achieved by two major classes of RL algorithms: actor-critic algorithms including direct heuristic dynamic programming (dHDP) [36, 37, 38, 39], and variants of policy iteration algorithms such as flexible policy iteration (FPI) [40]. Both types of single-agent based control algorithms were developed and demonstrated in simulated environments first. **Human Tests.** Then these algorithms were tested on human subjects walking with a robotic knee prosthesis [15, 16, 41]. Note that all of the above methodologies and tests used designer-prescribed robot joint movement profiles generated *a priori* for the specific subjects and walking tasks. In real-world use scenarios, joint movement profiles evolve dynamically in real-time to accommodate internal human walking objectives [14, 42]. A more recent single-agent RL control work [17] replaced designer-prescribed joint movement profiles with the intact joint motion. Note also that, none of the above results have directly taken into account human influence on human-robot walking in the control design, a novel contribution of this work. **Multi-agent reinforcement learning (MARL).**

A general cMARL problem usually cannot be transformed to an equivalent single-agent problem, as additional agent(s) and their control policies introduce uncertainties and/or non-stationarity to the environment [43]. Partial observability is common in cMARL problems, which are usually addressed by the decentralized POMDP framework. Centralized training decentralized execution (CTDE) is a popular paradigm to address multi-agent coordination problems. A central feature of most CTDE approaches is factorization of the joint state-action value function into individual utility functions. **MADDPG** [44] is a popular CTDE-based method to solve both cooperative and competitive MARL problems. **COMA** [45] is another popular MARL algorithm which utilizes a single centralized critic by using global state information and actions of all agents. We therefore use them in benchmark evaluations. **Shared Autonomy (SA).** The human-prosthesis problem under our consideration falls into the area of physical human-robot interaction (pHRI) as the human and the robotic prosthesis are physically coupled at all times. It is not, however, the extensively studied pHRI problem archetype (e.g., cooperative object manipulation, human operating in a remote environment), wherein interactions are usually mediated by a third object. For similar reasons, our pHRI problem is different in several important aspects from the approaches to the existing shared autonomy, such as cross-training [46], bounded memory adaptation [47], predict and blend [48], and model-free RL [49]. The wearable exoskeleton control problem seems relevant to our HPC problem. Yet, there are still fundamental differences. Recent exoskeleton approach also takes into account human-exoskeleton interacting effects [50, 51]. However, these exoskeletons mainly focus on end point performance of a foot or lower limb joints where user intent is quantified by estimating human joint torque or interactive torque. Thus, human-exoskeleton system performance goal is for the robot to produce a well-defined joint or endpoint trajectory [52, 53, 54]. For HPC problem we consider, there is no clear end point goal as there is in most existing current pHRI problems, including robotic upper limb that has been intensively addressed in literature. Therefore, how to define a performance goal for our HPC problem is a challenge. We address this study from existing limited yet proved knowledge [14, 42, 55]. **Modeling Challenges.** The HPC walking problem involves two strongly coupled agents, the interacting dynamics of which are difficult or nearly impossible to describe by ordinary difference or differential equations in large part because there is no clear performance goal in HPC as in studied cases of shared autonomy. HPC may be affected by several factors, such as lower limb mechanics, inter-limb neuromechemical coupling, and physical structure of the human body including the lower limb [56, 57]. Physical and physiological differences in individual human subjects further complicates modeling and control design. Even though our knowledge of human motor control and motor learning in cases such as post-stroke or general loss of normative locomotion capabilities have expanded greatly [58], little is known about how an active robotic prosthesis, not a traditional stick-type passive prosthesis, affects human and vice versa in walking tasks. This is because that the human-robotic prosthesis interacts continuously, the intricate human neurocontrol circuits including sensing, perception and feedback control, which are necessary to facilitate normative walking, is disrupted after amputation. **Human Utility Challenges.** Because of the reasons above, also because of no clear end point target as performance goals as in most studied shared autonomy cases, seamless collaboration between a robotic lower limb prosthesis serially attached to its human user poses new issues that requires to be answered [10, 11]. In this study, we based on latest understanding on the HPC problem, to provide an innovative solution approach to account for human influence in HPC.

**Contributions.** Human-robot collaborative tasks will continue to play an increasingly vital role in modern life, and existing single- and multi-agent control frameworks have only covered a small subset of the problems. The contributions of this work are as follows. 1) We introduce an innovative approach to automatically control a robotic lower limb prosthesis by treating the human user as a collaborating agent. We thus address a new challenge in the domain of shared autonomy problems. 2) To solve the control problem, we introduce a new cMARL approach to solve our HPC problem, a problem that cannot be readily solved to satisfaction by existing MARL approaches such as COMA or MADDPG. 3) This is the very first attempt in the field of wearable lower limb robots that human influence is explicitly considered in the robot control design.

## 3   Fundamental control problem statement

In powered lower limb prosthesis, the finite state impedance control (FS-IC) framework most frequently serves to provide intrinsic control of a robotic joint, i.e., it is the built-in controller from robotic prosthesis manufacturers. While position control is common in industrial robots, when a human is affixed to a robotic joint, robot trajectory tracking using position control may preclude any

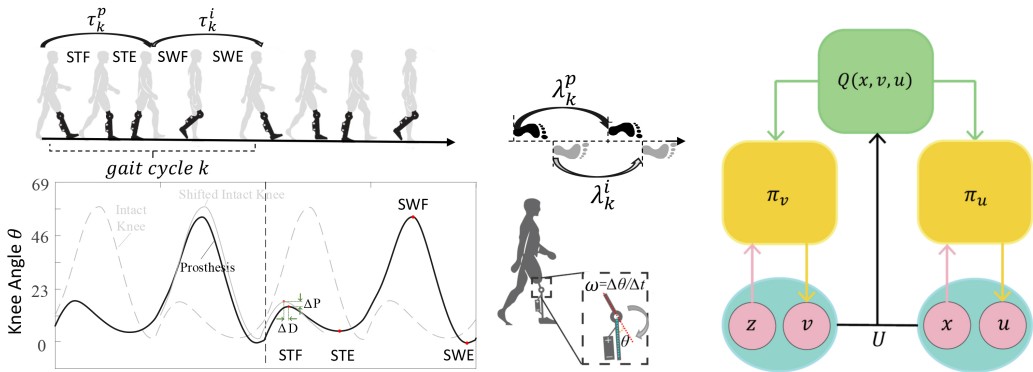

Figure 1: Left: human gait and prosthesis FS-IC characteristics. Right: cMARL solution approach to the HPC problem.

dynamic interaction of the robot with its human user and the environment [59]. This may cause an amputee to react to the awkwardness of the prosthesis rather than to interact constructively with it [60]. As FS-IC impedance control framework is considered to provide compliant control for human use of the robot, it is expected to generate stable and predictable human-robot walking behavior, and thus, it is adopted in this study which aims for its controller designs at real world applications.

The FS-IC treats a gait cycle, or a step, as four consecutive gait phases (Figure 1, Left): stance flexion (STF) as Phase 1, stance extension (STE) as Phase 2, swing flexion (SWF) as Phase 3, and swing extension (SWE) as Phase 4. In what follows, we design four individual controllers for each of the four phases. Since the four controllers are conceptualized similarly and designed using the same approach, for the sake of clarity we carry out subsequent discussion without explicitly referring to the specific phase numbers (but we emphasize that each of the four phases requires its own independently-trained controller). Additional details on FS-IC such as phase detection and transition are provided in Appendix A.3,

At the $k$-th gait cycle, a robot controller (solved by cMARL in this paper) is to determine three impedance parameters, $I_k = [K_k, B_k, (\theta_e)_k]^T \in \mathbb{R}^3$ under the FS-IC framework, representing stiffness $K_k$, damping coefficient $B_k$, and equilibrium position $(\theta_e)_k$, respectively. The prosthetic joint motor torque $T_k \in \mathbb{R}$ is then generated based on joint kinematics (knee joint angle $\theta$ and angular velocity $\omega$) according to the following impedance control law,

$$T_k = K_k(\theta - (\theta_{e_k})) + B_k\omega. \tag{1}$$

The control problem formulation requires automatically determining 12 control inputs or impedance control parameters (3 for each of the 4 gait phases) for individual users aiming at walk in real world situations. The initial set of feasible baseline impedance control parameters $I_0$ can be obtained from manufacturers or rehabilitation clinics. Impedance updates take place according to

$$I_{k+1} = I_k + u_k, \tag{2}$$

where $u_k \in \mathbb{R}^3$ is to be determined from our proposed cMARL approach.

## 4 Method

The human user and the robotic limb are considered collaborating agents. Our cMARL solution approach toward human-robot symmetrical walking problem is formulated based on physical features and measurements that have been shown affecting human-robot walking performance, and they are respectively available to the human and the robot, but not necessarily to each other.

**State and control variables.** Refer to Figure 1. At the $k$-th gait cycle, let $\tau_k^i$ and $\lambda_k^i$, respectively, represent the stance time (time of foot on the ground) and step length (length between two consecutive steps when toe touching the ground) of the human intact leg. Similarly, $\tau_k^p$ and $\lambda_k^p$, respectively, for the prosthetic leg. Let $\Delta\tau_k$, $\Delta\lambda_k$ denote respectively the difference of stance time and step length,

$$\Delta\tau_k = \tau_k^p - \tau_k^i, \ \Delta\lambda_k = \lambda_k^p - \lambda_k^i. \tag{3}$$

We thus define human state variable $z_k \in \mathbb{R}^2$ as $z_k = [\Delta\tau_k, \Delta\lambda_k]^T$.

Robot states include kinematic features determined from the knee movement profiles as well as step length and stance time (or $z_k$). For robot kinematic state variables, we extract gait kinematic features from knee motion profiles of both limbs (Figure 1). Let $P_k^i$ and $D_k^i$ represent the knee angle and time duration of the intact limb, and similarly, $P_k^p$ and $D_k^p$ for the prosthetic limb. Let $\Delta P_k$ and $\Delta D_k$ denote the error of peak knee angles and the error of duration time between the intact and the robotic knees, respectively,

$$\Delta P_k = P_k^p - P_k^i, \ \Delta D_k = D_k^p - D_k^i. \tag{4}$$

We then define robot state $x_k \in \mathbb{R}^4$ to include the following four variables, the first two of which are shared between the human and the robot, $x_k = [\Delta\tau_k, \Delta\lambda_k, \Delta P_k, \Delta D_k]^T$.

As such, human states are available to the robot, but a subset of the robot state (the kinematic features) is not available to the human. This reasonably reflects practical situations in real life applications.

The robot control policy is a state feedback law from robot state to robot impedance control parameters, namely, control $u_k$ consists of increments to the impedance control parameters (Eq. 2),

$$u_k = [\Delta K_k, \Delta B_k, (\Delta\theta_e)_k]^T. \tag{5}$$

Determining human control policy which originates from human neurocontrol circuit is a daunting task, especially now that sensing, perception and feedback control circuits are interrupted after a lower limb amputation. While agents' policy solutions are solved from Bellman optimality-like equations, a human cannot interpret and implement an MARL solution via their complicated neural circuits. But this is not our intention. We instead use the human control variable as an estimated input to the robot controller design, i.e., the robot is informed by an estimated human influence when they share the same goal of symmetrical walking.

As the very first step to demonstrate this idea, we consider the human user of the wearable robot intentionally or voluntarily walk at a reference step length, which may also be from an instructional feedback. For real life scenarios, such behavioral cues can change over time, task, or environment. But it is important to validate such formulation of considering human influence in the robot control design. Toward this end, we consider a practical reference cue denoted as a desired step length $\lambda_o$. We let $v_k$ represent an endogenous control signal of the human which is a function of human physical and mental states that involve activities ranging from neural level to muscular and joint level. Accordingly, we represent the endogenous human influence solved from the MARL design as a step length $\lambda_k^d$. We therefore define human control as,

$$v_k = \lambda_k^d - \lambda_o. \tag{6}$$

Including this estimated human control into the problem of robot control design, we take into account human influence on human-robot walking performance while they share the same symmetrical walking goal.

**Symmetrical walking as shared task goal.**

We consider the stage cost to be shared between the human and the robot.

$$U(x_k, v_k, u_k) = x_k^T R_x x_k + R_v v_k^2 + u_k^T R_u u_k + \mu h_k^2, \tag{7}$$

where $R_x \in \mathbb{R}^{4\times4}$ and $R_u \in \mathbb{R}^{3\times3}$ are positive semi-definite weighting matrices, $R_v$ and $\mu$ are positive weighting constants, and $h_k = v_k - \bar{v}_k$. In the above, the difference between actual human step length $\lambda_k^i$ and reference $\lambda_o$, denoted as $\bar{v}_k = \lambda_k^i - \lambda_o$, is considered practical and available.

In this formulation, the shared control objective is represented in two ways. First, the robot kinematic state variables in $x_k$ are to match the intact knee, i.e., the robotic joint angle profile is to match that of the intact joint. Additionally, a walking symmetry measure is directly considered by the differences in step length and stance time between human and robot, commonly used gait symmetry measures [61, 62, 63]. The human is assumed to perceive the gait symmetry but do not have access to or understand robot kinematic data (peak knee deflection error $\Delta P_k$ and peak knee time error $\Delta D_k$). The works [14, 64] show from test subject data that human control and adaptation has a direct influence on robot kinematics through dynamic interactions [59]. To account for this dynamic learning phenomenon, this framework penalizes the estimation error $h_k = v_k - \bar{v}_k = \lambda_k^d - \lambda_k^i$

between human perceived difference in step length. An unexpected or undesired human action will increase the cost represented by $h_k$, while a diminishing $h_k$ indicates that human input that originated from the neuromuscular system matches actual human performance. Minimal controller energy expenditures are also included in the cost structure.

The human-robot control objective to be solve by MARL as a function of the state variables and the the controls is formulated as an infinite horizon, discounted cost $Q(x_k, v_k, u_k) = \sum_{j=k}^{\infty} \gamma^{j-k} U(x_j, v_j, u_j)$ where $\gamma$ $(0 < \gamma < 1)$ is the discount factor for the infinite-horizon problem.

**Solutions to control actions.** Let $v_k = \pi_v(z_k)$ and $u_k = \pi_u(x_k)$, where $\pi_v$ and $\pi_u$ are the human and the robot control policies solved from cMARL, respectively. Solving optimal control policies $\pi_v^*$ and $\pi_u^*$ requires solving the optimal $Q$ function that satisfies Bellman optimality equation:

$$Q^*(x_k, v_k, u_k) = U(x_k, v_k, u_k) + \gamma Q^*(x_{k+1}, \pi_v^*(z_{k+1}), \pi_u^*(x_{k+1})). \tag{8}$$

When using a neural network-based actor-critic solution framework, we approximately solve the optimal control problem using the following iterative procedure ($i$ is the iteration index),

$$Q_{i+1}(x_k, v_k, u_k) = U(x_k, v_k, u_k) + \gamma Q_i(x_{k+1}, \pi_{v_i}(z_{k+1}), \pi_{u_i}(x_{k+1})). \tag{9}$$

where $\pi_{v_i}(z_k) = arg\min_{v_k} Q_i(x_k, v_k, u_k)$, and $\pi_{u_i}(x_k) = arg\min_{u_k} Q_i(x_k, v_k, u_k)$, and $Q_i(x_k, v_k, u_k)$ are iterative actor policies and iterative $Q$ value function.

During training, the actor and critic back-propagate their respective squared error to update their weights. The prediction error of actor $e_{a_v, k}, e_{a_u, k} \in \mathbb{R}$ is,

$$e_{a_v, k} = e_{a_u, k} = \frac{1}{2}(Q_i(x_k, v_k, u_k))^2. \tag{10}$$

The prediction error for the critic $e_{c,k}$ is formulated based on the Bellman error,

$$\epsilon_{c,k} = U + \gamma Q_i(x_{k+1}, \pi_{v_i}(z_{k+1}), \pi_{u_i}(x_{k+1})) - Q_{i+1}(x_k, v_k, u_k), \tag{11}$$

and the critic neural network is trained to minimize $e_{c,k} = \frac{1}{2}\epsilon_{c,k}^2$.

The optimal state-action cost-to-go function $Q^*(x_k, v_k, u_k)$ is approximated by a critic neural network which learns the $Q$ function by minimizing the Bellman error on the shared cost signal, not on a local cost signal for either the human or the robot as the human and the robot are physically coupled. We use our established direct heuristic dynamic programming (dHDP) algorithm [36] to solve this approximation dynamic programming problem. The critic neural network is a three-layer MLP with 6 hidden units and uses linear activation function in the output layer. Therefore, we have the approximated cost to go value represented by:

$$\hat{Q}_i(x_k, u_k) = W_{c2,i}\varphi\left(W_{c1,i}\left[x_k^T, u_k^T\right]^T\right), \tag{12}$$

where $W_{c1,i} \in \mathbb{R}^{6 \times 8}$ denotes the weight matrix between the input layer and the hidden layer, and $W_{c2,i} \in \mathbb{R}^{1 \times 6}$ the weight matrix between the hidden layer and the output layer during the $ith$ learning update. The weight updates of the hidden layer matrix $W_{c2}$ are according to

$$\Delta W_{c2,i} = l_c\left[-\frac{\partial e_{c,k}}{\partial W_{c2}}\right], \tag{13}$$

and the weight updates of the input layer matrix $W_{c1}$ are according to

$$\Delta W_{c1,i} = l_c\left[-\frac{\partial e_{c,k}}{\partial W_{c1}}\right], \tag{14}$$

where $l_c > 0$ is the learning rate of the critic network.

Similar to the critic network, the actor networks for the human and the robot, respectively are three-layer MLP with 6 hidden units with hyperbolic activation function in the output layer to bound the action output. The same SGD optimizer [36] can be applied to the actor networks as well.

The two actors, $u$ and $v$, respectively are

$$\begin{aligned} u_k &= \varphi\left(W_{a_u 2,i} * \varphi\left(W_{a_u 1,i} x_k\right)\right), \\ v_k &= \varphi\left(W_{a_v 2,i} * \varphi\left(W_{a_v 1,i} z_k\right)\right), \end{aligned} \tag{15}$$

where $W_{a_u1} \in \mathbb{R}^{6 \times 4}$, $W_{a_v1} \in \mathbb{R}^{6 \times 2}$, $W_{a_u2} \in \mathbb{R}^{3 \times 6}$ and $W_{a_v2} \in \mathbb{R}^{1 \times 6}$ are the weight matrices, and $\varphi(\cdot)$ is the hyperbolic tangent activation function used in the hidden layer and the output layer. The weight updates of the hidden layer matrix $W_{a2,i}$ are according to

$$
\begin{aligned}
\Delta W_{a_u2,i} &= l_a \left[ -\frac{\partial e_{a_u,k}}{\partial W_{a_u2,i}} \right], \\
\Delta W_{a_v2,i} &= l_a \left[ -\frac{\partial e_{a_v,k}}{\partial W_{a_v2,i}} \right].
\end{aligned}
\tag{16}
$$

The weight updates of the input layer matrix $W_{a1,i}$ are according to

$$
\begin{aligned}
\Delta W_{a_u1,i} &= l_a \left[ -\frac{\partial e_{a_u,k}}{\partial W_{a_u1,i}} \right], \\
\Delta W_{a_v1,i} &= l_a \left[ -\frac{\partial e_{a_v,k}}{\partial W_{a_v1,i}} \right],
\end{aligned}
\tag{17}
$$

where $l_a > 0$ is the learning rate of the actor.

Further details on the cMARL automatic control solution are provided in Appendix D.

## 5 Experiment

We conduct design evaluations of the proposed cMARL for symmetrical walking using OpenSim, a well-established open source biomechanical modeling tool for conducting biomechanics research and motor control science [65]. The robot knee control is realized within an FS-IC framework. In real life, initial impedance parameters can be selected based on manufacturer and/or rehabilitation clinician's recommendations. Similar care is given to OpenSim simulated walking yet in all evaluations, the initial impedance values vary from a large range of settings for fair examination. As in [37], we enforce realistic safety constraints to prevent the human from stumbling or falling. Impedance parameters are reset to initial impedance values if any of the state variables exceed the safety bounds. The safety protocols followed in this work are fully described in Appendix A.4.

To make the simulations reflective of real world conditions, sensor and actuator noise data extracted from real human experimental testing sessions is applied to all the simulations in this study. Appendix A.2 provides the complete procedure followed of extracting noise data from experiments involving human subjects and injecting it into all the simulations.

In this section, we provide results of a large set of simulation studies aiming at answering the following questions: 1) Does our cMARL solution framework provide better performance than state-of-the-art baselines, including MADDPG and COMA? 2) Does including human influence in robot control design accelerate learning and improve success rate of policy in comparison to single-agent based approach (wout/human)? 3) Is our cMARL applicable to different and realistic walking tasks? To provide answers, we show three sets of evaluations: benchmark, ablation and reliability. Benchmark and ablation evaluations are based on a level ground walking task with a pace of 1m/s. Reliability evaluations are based two new walking tasks: slope walking on a 11.5 degree ramp and level ground walking at an increased pace of 1.12m/s.

**Performance Criteria.** In order to ensure that amputee subjects walk safely and continuously, we consider several performance metrics: 1) As an optimal control problem, the objective is to minimize state regulation cost (smaller is better). 2) Peak knee error can directly reflect amputee safety, preventing falling and stumbling (smaller is better). 3) Symmetry in walking can prevent secondary injury (closer to 0 is better). 4) Fast learning in terms of fewer tuning steps is practically important to amputees (fewer steps is better). 5) High success rate boosts amputees' confidence and hence walking performance (higher is better). Details of how the data was obtained can be found in Appendix E.2. The main evaluation results are presented in Table 1. The second value in each entry (i.e., after the ± symbol) represents the standard deviation of the performance metrics.

**Benchmark study.** MADDPG and COMA differ from our proposed cMARL approach toward symmetry walking. MARL problems can vary greatly, same are expected of their solutions [66, 67].

To perform benchmark studies, we tailor MADDPG and COMA, respectively to our HPC problem. Details are provided in Appendix D.

Table 1: Performance of the implemented algorithms in terms of the five performance criteria. The results of the best-performing algorithm for each criterion are boldfaced

| Performance | w/human | wout/human | COMA | MADDPG |
|---|---|---|---|---|
| Stage Cost | $\mathbf{0.002 \pm 0.001}$ | $0.008 \pm 0.003$ | $0.004 \pm 0.002$ | $0.38 \pm 0.306$ |
| Peak Error | $\mathbf{0.003 \pm 0.001}$ | $0.007 \pm 0.002$ | $0.003 \pm 0.001$ | $0.025 \pm 0.014$ |
| Symmetry | $\mathbf{0.001 \pm 0.001}$ | $0.006 \pm 0.003$ | $0.003 \pm 0.002$ | $0.022 \pm 0.012$ |
| Converge Steps | $\mathbf{97 \pm 88.3}$ | $187.5 \pm 105.2$ | $136 \pm 121.8$ | – |
| Success Rate | $\mathbf{0.7}$ | $0.58$ | $0.5$ | – |

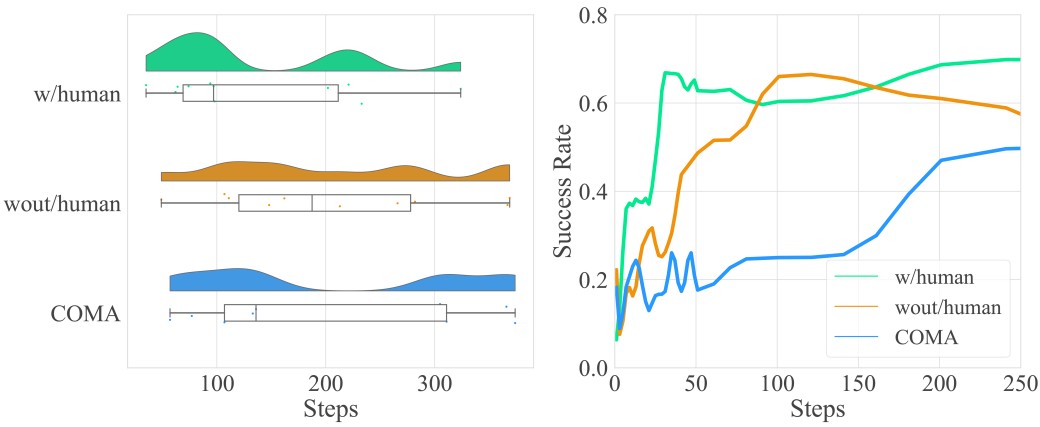

Figure 2: The total number of steps needed to reach convergence in training (left). The success rate during evaluation (right).

Figure 2 shows that our cMARL solution has the best convergence profile and success rate over the baselines. Further results on training and evaluation are shown in Figures 3 and 4 which compare the environment sample efficiency as well as algorithm performances. To answer the first question based on benchmarking, our cMARL solution outperforms the baselines both in terms of kinematic and symmetry measurements as shown in Figures 3 and 4, center and right panels for training and evaluation, respectively. Additionally, the left panels of Figures 2, 3 and 4 show that our cMARL solution outperforms benchmarks with at least 30% less environment samples.

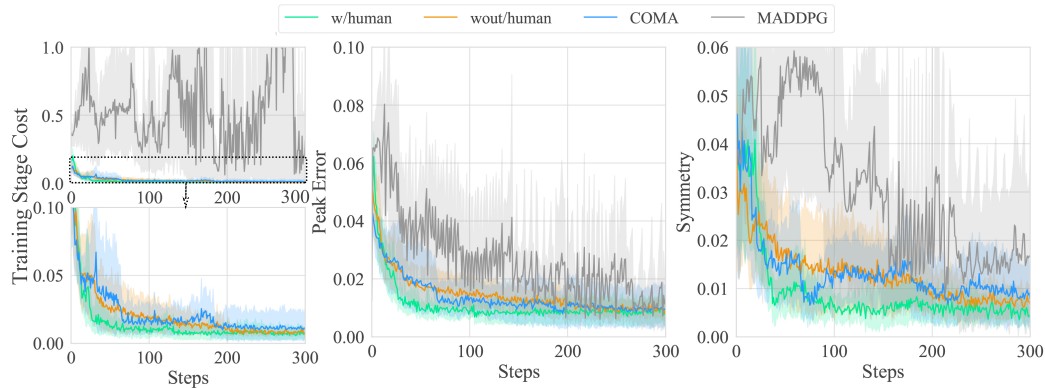

Figure 3: Learning curves of stage cost during training (left), peak angle error (middle) and symmetry of step length (right) for benchmark (w/human, COMA, MADDPG) and ablation (w/human, wout/human) studies. Each learning curve is averaged over 16 different random seeds and shaded by their respective 95% confidence interval.

**Ablation study.** To gain insights on how human control influences human-robot walking performance, an ablation study is carried out with direct human influenced terms removed, including $h_k$ and $v_k$,

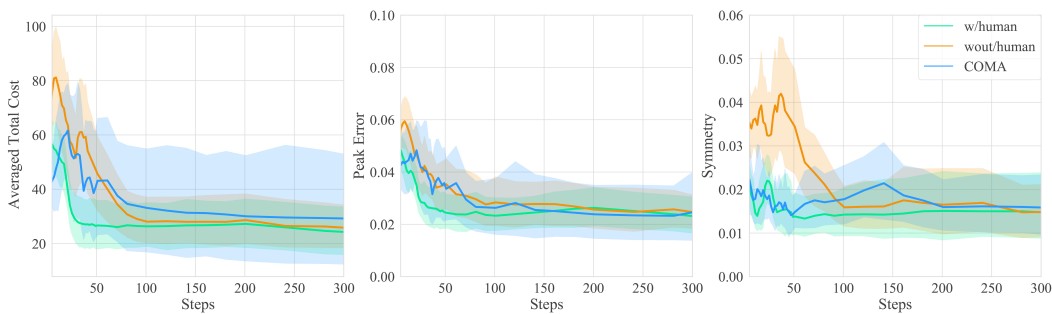

Figure 4: Learning curves of averaged total cost (terms associated with human influence removed) during evaluation (left), peak angle error (middle) and symmetry of step length (right) for benchmark (w/human, COMA) and ablation (w/human, wout/human) studies. Each (smoothed) learning curve is averaged over 5 different random seeds and shaded by their respective 95% confidence interval.

from the problem formulation. This only leaves the kinematics and symmetry measurements in the stage cost. Similar to the benchmark study, the ablation study has a training session and evaluation session. For comparable results, the total cost in evaluation is obtained as $U = (x)^T (R_x) x + u^T R_u u$. Figures 3 and 4 show that with an estimated human control influence accounted for in the robot control design, our cMARL solution (the green curves) outperforms the one without it (orange curves). Treating human and robot as collaborating agents toward a shared performance goal, our cMARL solution approach achieves increased success rate and accelerated learning speed (Figure 2). This result makes sense as an estimated human control provides a predictive signal to the robot control which aims at duplicating the intact human joint movement. Additional information on the quality of estimated human control is given in Appendix B.2.

**Reliability study.** To make the proposed cMARL method practical and useful in real life, we setup two new walking tasks: (1) slope walking (11.5 degree ramp) and (2) walking at an increased pace (1.12m/s). They will result in different walking patterns from those used in the baseline study, and thus different knee joint profiles. For slope walking, knee flexion will be more pronounced during the stance phase since it walks inclined. In the case of faster pace, stance time will be compressed. To carry out the tests, the same training procedure is used as in benchmark and ablation studies.

Figure. 5 shows the performance of cMARL during slope walking and increased pace walking tasks. Performance of the two new tasks follow the same trend as that of the level ground walking at a nominal pace. These results again validate our design approach of using the intact knee movement trajectory as the target for the robotic knee to copy. By doing so, we have removed a major control design barrier in the way of performing different walking tasks by automatic control.

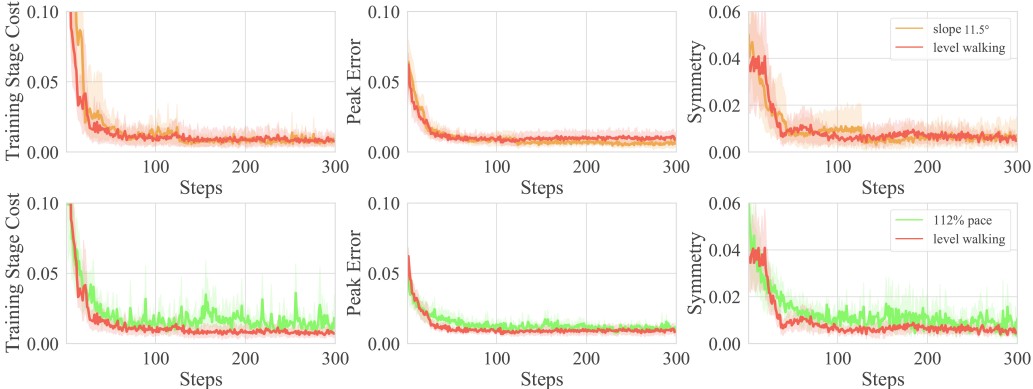

Figure 5: Learning curves of stage cost during training (Left), peak angle error (Middle) and symmetry of step length (Right) for different walking tasks (level ground walking at increased pace and ramp walking). Each learning curve is averaged over 16 different random seeds and shaded by their respective 95% confidence interval.

**Limitations of This Study.** Here we conduct an in-depth simulation-based analyses as an important first step to evaluate this novel cMARL solution to the HPC problem. Simulations are critically important as exploration of problem formulation, control algorithm design, and systematic evaluation are necessary to be performed prior to human experiment due to factors such as human fatigue, human safety, human loss of interest/confidence caused by repeated trial-and-error, time spent, and significant cost associated with testing amputee subjects. However, our framework is still to be tested in human experiment. Based on several important works in the literature [37, 40, 68, 69], extensive simulation studies followed by real life human test studies [41, 17, 70] have proven a highly successful development procedure for human/robot control. This will be the next step of this study.

For scenarios in which the terrain or task has changed significantly, a task planner will become necessary, making the problem a planning and control problem (as opposed to automatic control, the focus of this work). This expanded automatic control algorithm must be extensively verified in simulation and then in human tests before it can be integrated into a real-world planning framework for daily use cases. Such planning frameworks constitute intended future works.

## 6  Conclusion and discussion

1) In the US, approximately 1.7 million people live with limb loss. The amputee population is expected to double by 2050 as the population ages and incidence of dysvascular disease increases. As most lower limb amputees use prosthetic legs to restore basic bipedal locomotion, our solution to the prosthesis control problem can potentially help improve the function and quality of life of lower limb amputees. 2) In this work, we develop a novel cMARL framework towards systematically integrating the human and robot as collaborative agents to achieve normative walking toward solving real world problems. With reaching symmetric locomotion as shared control performance goal, we demonstrate improved walking performance. 3) Symmetry is selected as the shared goal for the collaborating agents because asymmetric walking has been linked to secondary health complications including back pain and osteoarthiritis [71, 72, 73]. Although human-robot walking performance goals are difficult to systematically catalogue, additional considerations such as embodiment of the robot into the human will be considered in future works. 4) By breaking apart the shared cost for the human and the robot (cf. Section 5.1), the symmetrical walking task is treated and evaluated by MADDPG and COMA, respectively. Simulation results show that the factorization-based CTDE paradigm struggles to address the human-robot problem. The observed performance issues with factorization likely stem from the intrinsic coupling between the human and robot agents. 5) While ensuring human user safety has been carefully considered during control design, additional analysis of important properties such as convergence of learning, (sub)optimality of control policy, and human-robot closed-loop stability is still needed for this framework. Encouragingly, previous related works [74, 75, 76, 77] indicate that these theoretical results are very likely to be provable.

## 7  Acknowledgments and Disclosure of Funding

This research was supported in part by NSF under grants 1563921, 1808752 and 2211740 for Si; grants 156454, 1808898 and 2211739 for Huang. Zhikai Yao participated in early stage discussion involving the formulation of shared performance goal.

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
