# Appendix A   Human-robot walking simulation using OpenSim

## A.1   OpenSim environment

OpenSim is a well-established open source biomechanical modeling tool for conducting biomechanics research and motor control science [65]. In this study, a five rigid-segments bipedal model including a pelvis, two thighs and two shanks is implemented on a rigid level platform to simulate level ground walking, up hill walking, and walking at a different (fast) pace. This simulation platform has been used in reports of previous single-agent based reinforcement learning control of the robotic knee. The simulation validated conceptualization of the solution approaches as well as the resulted controller structures have consequently been adopted in and adapted to human experiments.

The pelvis segment is linked to the ground platform using a slider joint, which allows the body to move relative to the ground. The thigh segments are linked to the pelvis using one-degree-of-freedom pin joints (hip joints). A pre-prescribed motion according to a well-established, normative data set [78] is applied to the hip joints. The shank segments are attached to the thighs using one-degree-of-freedom pin joints (knee) as well. Two torque actuators are applied to the knee joints for both sides so that the knee motion can be controlled by the torque. How torque control takes place via impedance controller and how impedance value updates are described in in appendix A.6. At the end of the shank segments, a contact sphere is set to simulate the ground reaction force between the foot and the ground using Hunt and Crossley contact force model. The range of hip joints is limited between [-100,100] degrees while the knee joint is limited in [-140,0] degrees to avoid over extension of the knee. Additional model settings, such as segment length, body mass and inertial parameters, are according to the lower limb OpenSim model in [79]. The left limb is designated as human controlled while the right limb as robot learning controlled [78]. Both knees are set to 0 degree at initialization and the initial angular velocities are -3.58 degree/s (right) and -3.32 degree/s (left), respectively.

In experimental evaluations, to simulate up hill walking, the ground is set as a 11.5 degree inclined ramp. To simulate walking at a different pace, the pre-prescribed hip motion is adjusted to allow a speed up to 112%. The rest of the settings are the same as the level walking task described above.

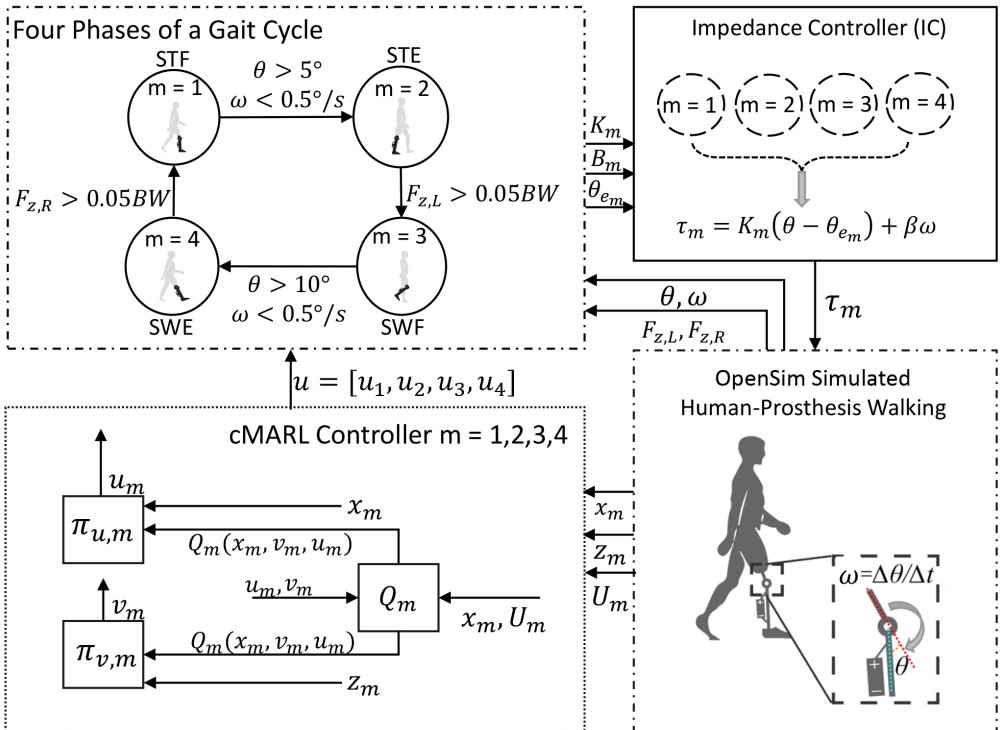

Figure 6: The OpenSim simulation platform of human-prosthesis walking with cMARL control.

## A.2 From real human experiment data to extract sensor noise and use them in simulations

To simulate state variables as if they were measured from experiments, we add noise to the simulated states. The sensor noise profiles were generated based on gait-to-gait variances of human-robot walking data captured from able-bodied subjects walking with prosthesis during stable and continuous walking of a single-agent based study. All experiments were approved by the Institutional Review Board. A total of 120 gait cycles of the knee joint movement trajectories were recorded in a level ground test during which the subject was required to walk wearing a prosthesis with a fixed impedance parameter after tuning. A goniometer was attached to measure the intact side knee motion. The prosthesis knee motion was directly read from the built-in sensors. The feature points (peak angle $P^p$, $P^i$ and phase duration $D^p$, $D^i$) of each gait cycle as shown in Figure 1 were collected. The standard deviations of intact and prosthetic knee feature points ($P^p$, $P^i$, $D^p$, $D^i$) were computed. The actuator noise was set at the same level as that of hardware instrumentation precision level which is 1% of the respective normative values of the impedance parameters.

The sensor and actuator noise are added as a white Gaussian noise with the standard deviation obtained from the above procedures. The standard deviations are 0.64 degree and 0.29 degree for $P^i$ and $P^p$, respectively. For $D^i$ and $D^p$, the standard deviations are 0.5% and 0.24%, respectively. The sensor noise is added to the state $x_k$ at each simulated step and the actuator noise is added to the action $v_k$ and $u_k$ at the output of the respective actor networks. The white noise added in each phase follows the same noise profile (i.e., the same Gaussian white noise) because the hardware does not change among phases.

## A.3 Finite State Impedance Control (FS-IC)

The FS-IC is the most employed intrinsic control framework for wearable lower limb prosthesis. Impedance control, also known as "compliance control", is well-established as a safe and reliable control strategy for lower limb prosthesis [59]. It allows a human to interact with a robot, rather than a position control that forces the amputee to react to the prosthesis. Almost all commercially available computer-controlled prostheses incorporate impedance control. In this study, impedance control of prosthesis joints mimics how humans control their biological joints in legs in walking. The finite state machine is used to mimic periodic gait cycles where each gait cycle is partitioned into four phases. For each phase, the prosthetic system mimics a passive spring-damper-system with predefined impedance [4, 5, 6] (i.e. stiffness $K$, damping coefficient $B$, and equilibrium position $\theta_e$) that matched the biological knee impedance.

Refer to Figure 6, a gait cycle is divided into four phases in the FS-IC: stance flexion (STF, $m = 1$), stance extension (STE, $m = 2$), swing flexion (SWF, $m = 3$) and swing extension (SWE, $m = 4$). The phase transitions are determined by knee motion and gait events (heel strike and toe-off) [7, 8] that are obtained from vertical ground reaction forces of both legs. According to Figure 6, the transition from STF to STE is made when the knee angular velocity $\omega$ is less than 0.5 degree per second and the knee angle $\theta$ greater than 5 degree. Similar condition is set for the transition from SWF to SWE as the angle threshold is set to 10 degrees. On the other hand, the transition between stance and swing is made according to the ground reaction force. Specifically, the transition from SWE to STF is triggered when the ground reaction force $F_{z,L}$ greater than a small threshold , 0.05 (5%) of Body Weight (BW), which is set to avoid false detection caused by noise. And the transition from STE to SWF is triggered when the contralateral limb hits the ground ($F_{z,R} > 0.05BW$).

For each of the four FS-IC phases $m$, there is a cMARL controller (Figure 6) to generate the respective control $u_m$ which is the adjustment to the current impedance values $I_m$ ([stiffness $K$, damping $B$, and equilibrium position $\theta_e$]). Therefore, 12 impedance parameters, 3 for each of the 4 phases, ($I = [I_1, I_2, I_3, I_4]$ ), are needed to simulate a gait in OpenSim where the prosthetic knee (right) is controlled by FS-IC with impedance setting $I$.

## A.4 Safety and reliability

In our real-world setting, safety and reliability for an integrated human-robot system are our consideration and Three safety features were implemented in the experiment to prevent "un-natural" control.

First and foremost, RL control is built on top of the finite state machine (FSM) impedance control , also known as "compliance control" (refer to Appendix A.3), which allows human to interact with robot, unlike position control that forces amputee to react to the prosthesis [60].

Next, safety bounds on knee angle and phase duration, step length and stance time are placed within FSM [40]. If a potential fall is detected, the RL controller is reset to the initial stabilizing controller. Specifically, as shown in Table 2, the safety bound is set at 1.5 times the standard deviations of the respective knee kinematic peak values observed in each phase [80]. If a state exceeds the safety bound, which means the prosthetic knee may place subjects into an unsafe situation, the impedance parameters of the prosthetic knee will be reset back to the initialized impedance values, but the actor and critic network weights are kept and will carry on with further training.

Lastly, actions generated from RL are bounded, a consideration which also reflects realistic constraints on actuation to limit control gain and avoid significant jump in impedance parameters.

Table 2: Safety bounds and tolerance bounds

| | Phase 1 | Phase 2 | Phase 3 | Phase 4 |
|---|---|---|---|---|
| Safety Bounds $\begin{bmatrix} Angle(°) & Duration(\%) \\ StepLength(m) & StanceTime(s) \end{bmatrix}$ | $\begin{bmatrix} 10.5 & 12 \\ 0.2 & 0.2 \end{bmatrix}$ | $\begin{bmatrix} 7.2 & 12 \\ 0.2 & 0.2 \end{bmatrix}$ | $\begin{bmatrix} 9 & 12 \\ 0.2 & 0.2 \end{bmatrix}$ | $\begin{bmatrix} 6 & 12 \\ 0.2 & 0.2 \end{bmatrix}$ |
| Tolerance Bounds $\begin{bmatrix} Angle(°) & Duration(\%) \\ StepLength(m) & StanceTime(s) \end{bmatrix}$ | $\begin{bmatrix} 1.5 & 2 \\ 0.02 & 0.02 \end{bmatrix}$ | $\begin{bmatrix} 1.5 & 2 \\ 0.02 & 0.02 \end{bmatrix}$ | $\begin{bmatrix} 1.5 & 2 \\ 0.02 & 0.02 \end{bmatrix}$ | $\begin{bmatrix} 1.5 & 2 \\ 0.02 & 0.02 \end{bmatrix}$ |

### A.5 Setting up human control prior to simulating human-robot as collaborating agents

To simulate an intact limb controlled by a human during walking, we design an impedance controller so that the simulated human-robot walks at a designated step length of $\lambda_0$ as in Eq. 6. This procedure is performed prior to cMARL control design implemented in OpenSim.

Both left and right knee controls are enabled by FS-IC in OpenSim. The right knee (prosthesis) is controlled by a fixed set of impedance parameters that correspond to a normative gait profile. This set of impedance parameters does not go through learning. The left knee (intact) control goes through learning and adaptation. A single agent reinforcement learning controller (without human involvement) is implemented. The state variable $s$ is defined as the difference between the actual step length and the desired step length, i.e., $s = \lambda^i - \lambda_0$. The action is still the same as in Eq. 5, i.e., $u = [\Delta K, \Delta B, (\Delta \theta_e)]^T$ .The stage cost is defined as $U = sR_s s + u_s R_u u_s$, where $R_s = 1$ and $R_u = diag(0.01, 0.01, 0.01)$. The tolerance bound for state variable $s$ is $\pm 0.015$ from the desired step length $\lambda_0 = 0.75$.

Three training trials, each with 300 steps, with different random seeds are used to sequentially update the intact knee control. Namely, the impedance parameters are kept from one trial transitioning into the next but the network weights are randomly initialized to enable exploration. This procedure has resulted in a reliable human controlled knee. With this controller, the simulated human walking is then enabled by this set of impedance parameters, which results in a step length of 0.75. This set of impedance control parameters remain unchanged to carry out the study.

### A.6 Integrating cMARL controller into FS-IC and OpenSim simulation of walking

Figure 6 is the overall structure based on which we have performed the evaluations. First of all, the bipedal model OpenSim environment is initialized according to Appendix A.1. Secondly, the knee joint torque actuator is enabled by an FS-IC controller which parses a gait cycle into 4 phases according to Appendix A.3. A random but stabilizing initial set of impedance parameters is given to the FS-IC to enable walking. Then, a cMARL controller is initialized for each of four phases.

Once initialized, OpenSim simulation proceeds by passing the state $x_m$ and stage cost $U_m$ the cMARL, which triggers learning as shown in Algorithm in Appendix D. The control actions $u_m$ for each of the four phases, with the same form as in Eq. 5 but from respectively different cMARL controllers, are placed into the FS-IC. Then FS-IC will update the impedance parameters by Eq. 2 and provide continuous torques based on Eq. 1 to the knee actuator. Finally, OpenSim simulated

gait provides the state $x_m$ and stage cost $U_m$ to cMARL. The above procedure repeats until cMARL learning terminates by reaching into the state $x_m$ the tolerance bound in Table 1.

## Appendix B   Additional Results

### B.1   Batch v.s. online implementation

In most of deep reinforcement learning applications such as humanoid in MuJoCo and many others, researchers usually train reinforcement learning modules in millions of steps and with memory size at a million level as well. Batch training with batch size of hundreds or thousands have been very effective, efficient and successful. However, in real world application like our human-robot cMARL problem, it is unrealistic to ask an amputee subject to walk millions of steps wearing a prosthesis. We therefore need to perform training in several gait cycles at the level of hundreds. As such, using batch training in our control framework, especially with in FS-IC, may not result in anticipated outcome as in deep neural network training. We therefore performed all experiments in this study using online learning. However, we have also performed an evaluation of training using batch vs. online modes. Figure (7) shows training performance of an implementation using a batch size of 3 and an online implementation. Comparisons are under the same comparable conditions including initial conditions. Results are based on 6 consecutive trails, each having 300 steps. Online learning converges faster than batch learning most likely due to that the network weights are updated more frequently.

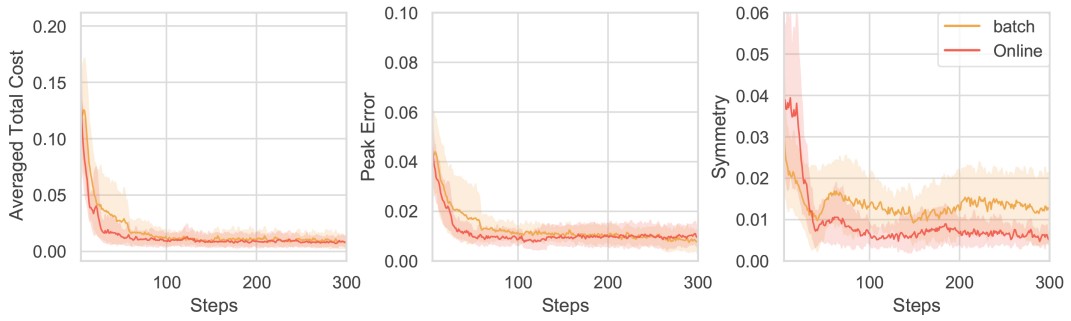

Figure 7: A comparison of online vs batch (batch size 3) training

### B.2   About estimated human control

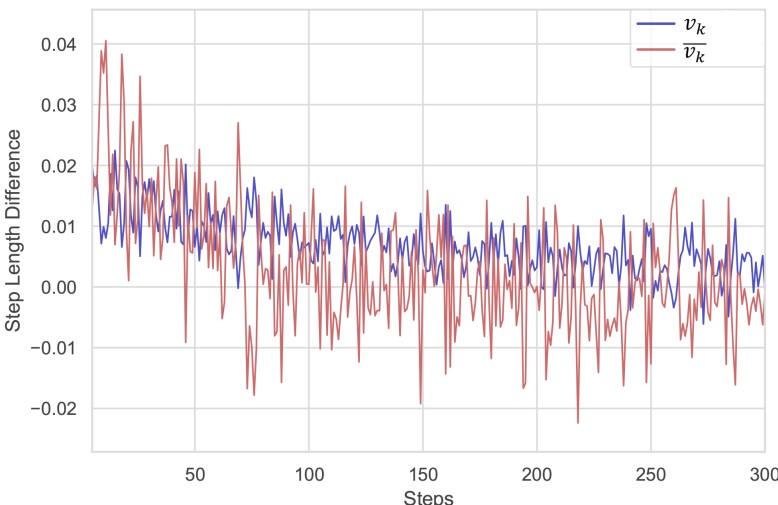

Figure 8: A comparison between the estimated human control $v_k$ and the actual human control $\bar{v}_k$ during evaluation.

We make use of an estimated human control $v_k$ in the design of robot control, all under a cMARL problem construct of shared control objective between a human and a robot. It is therefore important to know if the estimated human control $v_k$ is able to capture human action. Figure 8 shows that the estimated human action $v_k$ and the actual measured action $\bar{v}_k$ of an evaluation trial, based on a fully trained policy. It indicates that the estimated human action $v_k$ learns to approach actual human action $\bar{v}_k$. Moreover, $v_k$ fluctuates less than the actual, a phenomenon which suggests that $v_k$ may have become an inference bias of the human through learning to provide foresight on the ultimate goal, i.e. to optimize the shared objective function of cMARL by human and robot.

## B.3 Converged impedance parameters

| task | STF | | | STE | | | SWF | | | SWE | | |
|---|---|---|---|---|---|---|---|---|---|---|---|---|
| | $K$ | $B$ | $\theta_e$ | $K$ | $B$ | $\theta_e$ | $K$ | $B$ | $\theta_e$ | $K$ | $B$ | $\theta_e$ |
| level | 2.09 | 0.10 | -0.27 | 0.21 | 0.12 | -0.02 | 0.042 | 0.0055 | -1.03 | 0.055 | 0.0054 | -0.24 |
| level | 2.00 | 0.09 | -0.31 | 0.18 | 0.11 | -0.03 | 0.053 | 0.0059 | -1.05 | 0.060 | 0.0058 | -0.26 |
| level | 2.12 | 0.12 | -0.27 | 0.17 | 0.09 | -0.22 | 0.034 | 0.0049 | -1.02 | 0.056 | 0.0061 | -0.21 |
| level | 2.40 | 0.12 | -0.31 | 0.22 | 0.14 | -0.05 | 0.086 | 0.0084 | -0.90 | 0.074 | 0.0076 | -0.21 |
| level | 1.77 | 0.10 | -0.29 | 0.21 | 0.11 | -0.03 | 0.037 | 0.0043 | -1.00 | 0.047 | 0.0052 | -0.26 |
| slope | 1.48 | 0.12 | -0.26 | 0.16 | 0.10 | -0.13 | 0.042 | 0.0069 | -1.02 | 0.046 | 0.0048 | -0.26 |
| slope | 1.79 | 0.24 | -0.28 | 0.24 | 0.21 | 0.11 | 0.049 | 0.0072 | -1.10 | 0.048 | 0.0071 | -0.06 |
| slope | 2.48 | 0.13 | -0.29 | 0.19 | 0.09 | -0.09 | 0.044 | 0.0059 | -1.04 | 0.036 | 0.0045 | -0.24 |
| slope | 2.08 | 0.10 | -0.30 | 0.23 | 0.07 | -0.23 | 0.041 | 0.0066 | -1.03 | 0.044 | 0.0047 | -0.28 |
| slope | 1.97 | 0.10 | -0.21 | 0.19 | 0.02 | -0.04 | 0.088 | 0.0067 | -0.84 | 0.146 | 0.0210 | -0.03 |
| Pace | 2.04 | 0.10 | -0.32 | 0.18 | 0.11 | -0.06 | 0.013 | 0.0039 | -1.07 | 0.055 | 0.0051 | -0.30 |
| Pace | 2.44 | 0.10 | -0.31 | 0.17 | 0.10 | -0.14 | 0.033 | 0.0036 | -0.75 | 0.046 | 0.0057 | -0.27 |
| Pace | 2.18 | 0.13 | -0.33 | 0.17 | 0.08 | -0.05 | 0.044 | 0.0067 | -1.06 | 0.056 | 0.0058 | -0.26 |
| Pace | 1.49 | 0.13 | -0.36 | 0.15 | 0.09 | -0.09 | 0.044 | 0.0062 | -1.12 | 0.037 | 0.0049 | -0.21 |
| Pace | 2.15 | 0.10 | -0.29 | 0.15 | 0.07 | -0.20 | 0.032 | 0.0051 | -1.04 | 0.045 | 0.0060 | -0.19 |

Table 3: This table shows the final converged impedance parameters for all 4 phases from three different tasks described in 5

Table 3 summarizes all 12 impedance control parameters. Note however, such comparison may not yield much insight due to redundancy in human joint actuation space, and also strong coupling among parameters. On a side note, the current state-of-the-art practice in clinics is to manually tune 1 impedance at a time, a practice that cannot take into account of coupling, nor it is possible to tune all 12 parameters simultaneously best achieving a tuning goal.

## Appendix C   Simulated human environment

### C.1   Setting up Matlab scripting environment

Although OpenSim has a dedicated GUI interface, in this study we are using the OpenSim API to implement the environment in Matlab. Please follow the instructions provided by SimTK with the link below :https://simtk-confluence.stanford.edu:8443/display/OpenSim/Scripting+with+Matlab

Here are the key steps of configuration
1. Launch MATLAB in administrator mode.
2. Run the configureOpenSim.m under OpenSim/script directory
3. Choose the installation path of OpenSim in the dialog
4. Restart Matlab
5. Load OpenSim library by command: Import org.opensim.modeling.*

### C.2   Bipedal walking model building

The walking model was based on a bipedal model provided by SimTK. The model consists of a rigid platform and a four-link walker. The platform is connected to the ground by a pin joint that permits rotation about the Z-axis. The pelvis of the walker is connected to the platform by a FreeJoint object, which allows 6-degree-of-freedom (i.e., unconstrained) motion between the pelvis and the platform.PinJoint objects are used to connect the thighs to the pelvis and the shanks to the thighs. Two torque actuators are applied to the knee joints for both sides so that the knee motion can be controlled by the torque. At the end of the shank segments, a contact sphere is set to simulate the ground reaction force between the foot and the ground using Hunt and Crossley contact force model. The range of

hip joints is limited between [-100,100] degrees while the knee joint is limited in [-140,0] degrees to avoid over extension of the knee. Additional model settings, such as segment length, body mass and inertial parameters, can be configured through the .osim model configuration file in a xml format.

In this study, the left limb is designated as human controlled while the right limb as robot learning controlled. The torque actuator was added under tag <forceSet>/<objects>/< TorqueActuator>.

The original osim file can be downloaded from the link below.

https://simtk-confluence.stanford.edu:8443/display/OpenSim33/

From+the+Ground+Up %3A+Building+a+Passive+Dynamic+Walker+Model

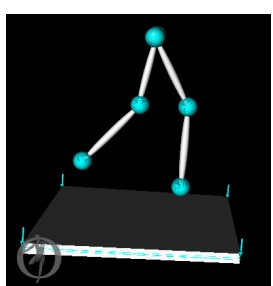

Figure 9: OpenSim human model

### C.3 Scripting OpenSim with Matlab

During the implementation, the following scripts was implemented as described below. The example file can be downloaded from the link: https://simtk-confluence.stanford.edu:8443/pages/viewpage.action?pageId=28777060

We customized some scripts to realize the desired function. We attached the modified code at the end of the appendix to help replicate the simulation environment. Table C.3 shows the scripts' name and their description to help reader understand their functionalities.

| Script Name | Description |
|---|---|
| IntergateQpenSimPlant.m | This function runs (and optionally visualizes) a forward simulation using one of Matlab's integrators. The script creates a plant function which returns state derivatives given a model and state values. This plant function can then be passed to any of the built-in integration tools in Matlab. The function IntegrateQpenSimPlant allows you to quickly run a simulation from the default values of the model in a single line. This function requires OpenSimPlantFunction. |
| OpenSimPlantControls Function.m | The function calculates a set of control values which are input to OpenSim muscles and actuators. |
| OpenSimPlantFunction.m | This function creates an interface which calculates the state derivatives for an OpenSim Model object and a OpenSim State object. This function can be passed to a Matlab integrator such as ode15s. |

### C.4 Structure of simulation environment

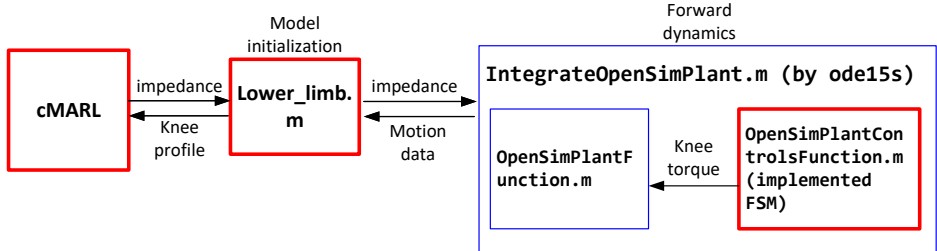

Figure 10: The OpenSim simulation platform of human-prosthesis walking with cMARL control.

Figure 10 shows the code structure of the simulation. The customized script will displayed in red box while the API script was in blue box. Lower_limb.m is the interface between Agent and the OpenSim. Agent feeds the updated impedance parameters through lower_limb and obtained the

knee profile as feedback. Lower_limb.m will call the OpenSim API by a ODE solver (ode15s) to compute the forward dynamics of the human locomotion. The communication rate between Agent and Lower_limb is once per gait cycle. And the communication rate between lower_limb and OpenSim is decided by ode15s solver. In the OpenSim, OpenSimPlantFunction.m will execute the customized function OpenSimPlantControlsFunction.m which contains the FSM-IC controller. All the customized codes were shared at the end of the Appendix.

## Appendix D    Baseline and ablation study implementation

In this study we compare cMARL with DDPG, COMA and cMARL without human agent (dHDP).

The code of DDPG and COMA was modified based on the public code to fit into our environment. By loading the corresponding class of algorithm in the main.m line 14, user can choose the different formation of cost and learning rule. Or customized by themselves. The example code of the controller will be released (with a URL to the github repository) along with the camera-ready version.

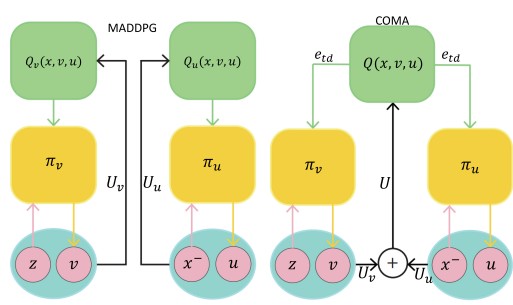

Figure 11: MADDPG (left) and COMA (right) tailored to solve the human-robot symmetrical walking problem.

For MADDPG, we allow human and robot have different stage cost as $U_v = z^T R_z z + R_v v^2 + \mu h^2$ and $U_u = (x^-)^T (R_{x^-}) x^- + u^T R_u u$, respectively where $x^-$ only includes the robotic joint kinematic variables. These stage costs respectively go to update critic $Q_v$ and $Q_u$ $Q_v = \sum_{j=k}^{\infty} \gamma^{j-k} U_v$ and $Q_u = \sum_{j=k}^{\infty} \gamma^{j-k} U_u$ for human and robot. Human policy ($\pi_v$) (function of $z$) and robot policy ($\pi_u$) (function of $x^-$) are determined accordingly. using $Q_v$ and $Q_u$ functions, respectively. Respective policies will execute based on their local observation $z$ and $x^-$. For COMA, we train a single centralized critic $Q = \sum_{j=k}^{\infty} \gamma^{j-k} U$ with joint stage cost $U = U_v + U_u$ where $U_v, U_u$ are the same as in MADDPG. Policies will execute locally. We update the policies using respective TD errors in place of a non-biased advantage function [45] because we are dealing with a strongly coupled human-robot system. Therefore it does not make sense to implement the counterfactual advantage function as in COMA. Also note that, we use real-time training instead of batch memory because collecting batch data for policy update requires an amputee subject to endure multiple gait cycles before one learning update. This may not be realistic for practical use. Nonetheless, an evaluation is performed (Appendix B.1). Details on the derivations and implementations of the tailored MADDPG and COMA are in Appendix D.

### D.1    cMARL without human agent

In the ablation study, we remove the human agent to study how effective of the human agent in the learning process. This turns the problem into a single agent approach (dHDP) [36].

**Learning Algorithm.** We consider the stage cost as

$$U(x_k, u_k) = x_k^T R_x x_k + u_k^T R_u u_k, \tag{18}$$

where $R_x \in \mathbb{R}^{4 \times 4}$ and $R_u \in \mathbb{R}^{3 \times 3}$ are positive semi-definite weighting matrices.

Solving optimal control policies $\pi_v^*$ and $\pi_u^*$ requires solving the optimal $Q$ function that satisfies Bellman optimality equation:

$$Q^*(x_k, u_k) = U(x_k, u_k) + \gamma Q^*(x_{k+1}, \pi_u^*(x_{k+1})). \tag{19}$$

When using a neural network-based actor-critic solution framework, we approximately solve the optimal control problem using the following iterative procedure ($i$ is the iteration index),

$$Q_{i+1}(x_k, u_k) = U(x_k, u_k) + \gamma Q_i(x_{k+1}, \pi_{u_i}(x_{k+1})), \tag{20}$$

where $\pi_{u_i}(x_k) = arg \min_{u_k} Q_i(x_k, v_k, u_k),$

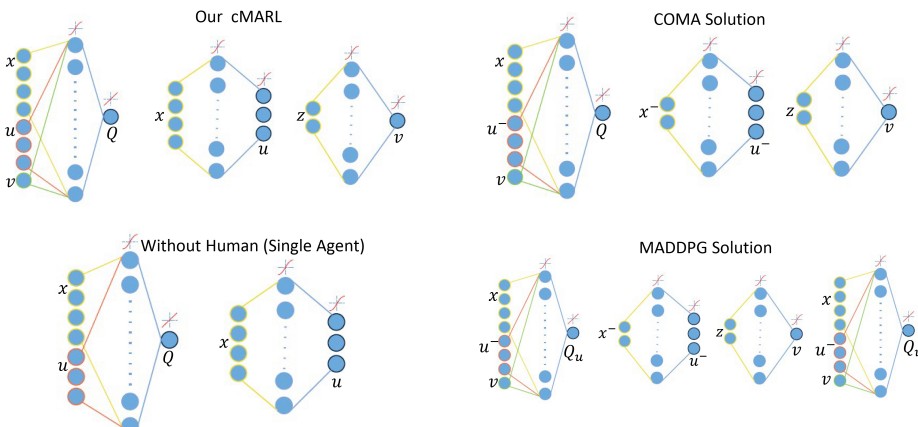

Figure 12: Actor-critic networks for benchmark evaluation and ablation evaluation. Clockwise: our proposed cMARL, COMA solution, MADDPG solution, and our approach without considering human or single-agent approach in the ablation study.

and $Q_i(x_k, u_k)$ are iterative actor policies and iterative $Q$ value function.

During training, the actor and critic back-propagate their respective squared error to update their weights. The prediction error of actor $e_{a_u,k} \in \mathbb{R}$ is,

$$e_{a_u,k} = \frac{1}{2}(Q_i(x_k, u_k))^2. \tag{21}$$

The prediction error for the critic $e_{c,k}$ is formulated based on the Bellman error,

$$\epsilon_{c,k} = U + \gamma Q_i(x_{k+1}, \pi_{u_i}(x_{k+1})) - Q_{i+1}(x_k, u_k), \tag{22}$$

and the critic neural network is trained to minimize $e_{c,k} = \frac{1}{2}\epsilon_{c,k}^2$.

**Network architecture.** For the ablation study, a single agent based on dHDP with one critic network and one actor network was implemented. To represent our neural shape, we use a 3-layer MLP with 6 hidden units for all networks. The critic network uses linear activation function in the output layer while the actor network uses hyperbolic function to bound the action output. The network was optimized by our own optimizer that based on stochastic gradient descent with back-propagation.

**Training parameters.** Since training speed is critical in a real human experiment, when optimizing the single agent, a learning rate of $\eta = 1 \times 10^{-2}$ was used to train both critic and actor network. The optimization target error was $\epsilon = 1 \times 10^{-3}$. The discount factor $\gamma = 0.95$ was used for all scenarios and ablation study. We initialize the network weights uniformly in the range [-1,1]. we set the weighting matrices in the stage cost Eq. 1 to be: $R_x = diag(1, 1, 0.25, 0.25)$, $R_u = diag(0.1, 0.1, 0.1)$, and $R_v = 0.1$.

The convergence of criteria was set as 1.5 degree for knee angle error and 2% for duration error which was decided by the device sensor accuracy in the real life and normal human noise. The observation and action noise were added to the simulation experiment. The detail of the human noise is mentioned in Appendix A.2.

Each training trail contains 300 gait cycles. Evaluations are conducted every 2 gaits during the first 50 gaits and then every 10 gaits afterwards, each using the latest policy at the time of evaluation.

## D.2 cMARL

Our cMARL method has been reported in the paper with its actor-critic network prediction errors described in Eqs. 10 and 11. During training, the actor and critic networks back-propagate their respective squared errors to update their weights.

**Learning Algorithm.** We consider the stage cost to be shared between the human and the robot.

$$U(x_k, v_k, u_k) = x_k^T R_x x_k + R_v v_k^2 + u_k^T R_u u_k + \mu h_k^2, \tag{23}$$

where $R_x \in \mathbb{R}^{4 \times 4}$ and $R_u \in \mathbb{R}^{3 \times 3}$ are positive semi-definite weighting matrices, $R_v$ and $\mu$ are positive weighting constants, and $h_k = v_k - \bar{v}_k$. In the above, the difference between actual human step length $\lambda_k^i$ and reference $\lambda_o$, denoted as $\bar{v}_k = \lambda_k^i - \lambda_o$, is considered practical and available.

Solving optimal control policies $\pi_v^*$ and $\pi_u^*$ requires solving the optimal $Q$ function that satisfies Bellman optimality equation:

$$Q^*(x_k, v_k, u_k) = U(x_k, v_k, u_k) + \gamma Q^*(x_{k+1}, \pi_v^*(z_{k+1}), \pi_u^*(x_{k+1})). \tag{24}$$

When using a neural network-based actor-critic solution framework, we approximately solve the optimal control problem using the following iterative procedure ($i$ is the iteration index),

$$Q_{i+1}(x_k, v_k, u_k) = U(x_k, v_k, u_k) + \gamma Q_i(x_{k+1}, \pi_{v_i}(z_{k+1}), \pi_{u_i}(x_{k+1})), \tag{25}$$

where $\pi_{v_i}(z_k) = arg\min_{v_k} Q_i(x_k, v_k, u_k), \pi_{u_i}(x_k) = arg\min_{u_k} Q_i(x_k, v_k, u_k)$, and $Q_i(x_k, v_k, u_k)$ are iterative actor policies and iterative $Q$ value function.

During training, the actor and critic back-propagate their respective squared error to update their weights. The prediction error of actor $e_{a_v,k}, e_{a_u,k} \in \mathbb{R}$ is,

$$e_{a_v,k} = e_{a_u,k} = \frac{1}{2}(Q_i(x_k, v_k, u_k))^2. \tag{26}$$

The prediction error for the critic $e_{c,k}$ is formulated based on the Bellman error,

$$\epsilon_{c,k} = U + \gamma Q_i(x_{k+1}, \pi_{v_i}(z_{k+1}), \pi_{u_i}(x_{k+1})) - Q_{i+1}(x_k, v_k, u_k), \tag{27}$$

and the critic neural network is trained to minimize $e_{c,k} = \frac{1}{2}\epsilon_{c,k}^2$.

**Network architecture.** As figure shows, cMARL contains one critic network and two actor networks. Same as single agent, we use a 3-layer MLP with 6 hidden units for all networks. The actor-critic code was built upon dHDP code. The critic network uses linear activation function in the output layer while the actor network uses hyperbolic function to bound the action output. The stage cost was defined in Eq. 6. The corresponding error functions were defined in Eq.9 and Eq.10.

**Training parameters.** Due to the training speed is critical in the real human experiment, when optimizing the cMAR, a learning rate of $\eta = 1 \times 10^{-2}$ was used to train both critic and actor network. The optimization target error was $\epsilon = 1 \times 10^{-3}$. The discounted factor $\gamma = 0.95$ .The convergence criteria are same as single agent dHDP.

### D.3 COMA

COMA is popular MARL algorithm which utilizes a single centralized critic by using global state information and actions of all agents. Additionally, COMA addresses the multi-agent credit assignment challenge by using a counterfactual baseline which keeps all other agents' actions fixed while treating a single marginalized agent. In game-playing problems, COMA significantly improves performance over other MARL methods such as independent Q-learning [81], multiagent bidirectionally-coordinated nets [82], and variant DQN [83]. However, due to the heavy dynamic coupling between the human and prosthesis, it is impossible to keep one agent action fixed and marginalize out the other agent's action. As a result, this counterfactual baseline is not appropriate for our problem, either. Our Mathlab version of COMA is based on [45] and weights updates using our own optimizer on E.6

**Learning Algorithm.** The original COMA uses a single centralized critic based on global states and actions. Additionally, COMA addresses the challenge of multi-agent credit assignment problem by using a counterfactual baseline that marginalises out a single agent's action while keeping other agents' actions fixed. However, in our cMARL human-robot symmetrical walking problem, human and prosthesis are tightly coupled both in time and space. It is impossible to break apart the two agents, i.e., it is not suitable for an implementation using counterfactual function. Therefore, we keep the single centralized critic but instead of using counterfactual advantage function, we use a TD error as a non-biased advantage function while each agent acts locally. The stage cost is represented by

$$U = U_v + U_u, \tag{28}$$

where $U_v$ and $U_u$ have the same structure as those in the above MADDPG solution. Additionally, our COMA implementation executes locally based on local observations. The centralized critic prediction error is

$$\epsilon_{c,k} = U + \gamma Q_i(x_{k+1}, \pi_{v_i}(z_{k+1}), \pi_{u_i}(x_{k+1}^-)) - Q_{i+1}(x_k, v_k, u_k^-), \tag{29}$$

and the critic neural network is trained to minimize $e_{c,k} = \frac{1}{2}\epsilon_{c,k}^2$. Additionally, the prediction error of actor for human $e_{a_v,k}$ and prostheses $e_{a_u,k}$ shared the TD error with the critic,

$$e_{a_v,k} = e_{a_u,k} = e_{c,k}, \tag{30}$$

where during training, the actor and critic back-propagate their respective squared error to update their weights.

**Network architecture.** As figure 12 shows, COMA shared the same architecture from cMARL. Our code is modified upon the public COMA code to fit in our environment. To fair comparison, the shape of network which are 3-layer MLP. The stage cost was defined in Eq. 11. The corresponding error functions were defined in Eq. 12 and 13.

**Training parameters.** To fair comparison, COMA use the same hyperparameter as cMARL which the learning rate of $\eta = 1 \times 10^{-2}$ was used to train both critic and actor network. The optimization target error was $\epsilon = 1 \times 10^{-3}$. The discounted factor $\gamma = 0.95$. The convergence criteria are same as cMARL.

### D.4 MADDPG

MADDOPG is another popular CTDE-based method to solve both cooperative and competitive MARL problems. Agents are assumed to communicate complete and perfect information with each other, so a centralized critic is trained from global state/action data, allowing agents to have individual reward functions. However, since it is unrealistic for the human to have direct access to robot kinematic sensor data, this structure is not immediately appropriate for the human-robot collaborative walking problem. Our Mathlab version of MADDPG is based on [44] and weights updates using our own optimizer on E.6

**Learning Algorithm.** The MADDPG was proposed to solve both cooperative and competitive learning problems by using different reward structures for each agent. Therefore, during training, each agent learns a centralized critic based on observations and actions of all agents. But for execution, each agent acts locally. In order to implement MADDPG, we make the human and the robot only use their local information, respectively. Specifically, we separate state $x_k$ into $z_k$ and $x_k^- = [\Delta P_k, \Delta D_k]^T$, signifying local observations for human and prosthesis, respectively. Therefore, the local actors $\pi_v$ and $\pi_u$ take local observations to generate actions $v_k$ and $u_k^-$, respectively. Additionally we formulate stage costs for human $U_v$ and prosthesis $U_u$, respectively as

$$
\begin{aligned}
U_v &= z_k^T R_z z_k + R_v v_k^2 + \mu h_k^2, \\
U_u &= (x_k^-)^T (R_{x^-}) x_k^- + u_k^{-T} R_u u_k^-,
\end{aligned}
\tag{31}
$$

where $R_v$ and $R_u$ have the same weights as our cMARL method, $R_{x^-} = diag(1,1)$ only takes the respective weights for prosthesis in $R_x$, and $R_z = diag(0.25, 0.25)$ which also, only extracts the weights for human in $R_x$.

During training, the actor and critic back-propagate their respective squared errors below to update their weights.

The prediction errors for human actor $e_{a_v,k}$ and prostheses actor $e_{a_u,k}$, respectively are

$$
\begin{aligned}
e_{a_v,k} &= \frac{1}{2}(Q_{v,i}\left(x_k, v_k, u_k^-\right))^2, \\
e_{a_u,k} &= \frac{1}{2}(Q_{u,i}\left(x_k, v_k, u_k^-\right))^2,
\end{aligned}
\tag{32}
$$

where $Q_{v,i}$ and $Q_{u,i}$ are respectively the $Q$ values of human and prosthesis during the $i$th iteration.

The prediction errors for the human critic $Q_v$ and prosthesis critic $Q_u$ are formulated based on their respective Bellman-like errors,

$$
\begin{aligned}
\epsilon_{v,k} &= U_v + \gamma Q_{v,i}(x_{k+1}, \pi_{v_i}(z_{k+1}), \pi_{u_i}(x_{k+1}^-)) - Q_{v,i+1}\left(x_k, v_k, u_k^-\right), \\
\epsilon_{u,k} &= U_u + \gamma Q_{u,i}(x_{k+1}, \pi_{v_i}(z_{k+1}), \pi_{u_i}(x_{k+1}^-)) - Q_{u,i+1}\left(x_k, v_k, u_k^-\right).
\end{aligned}
\tag{33}
$$

The critic neural networks $Q_v$ and $Q_u$ are trained to minimize $e_{v,k} = \frac{1}{2}\epsilon_{v,k}^2$ and $e_{u,k} = \frac{1}{2}\epsilon_{u,k}^2$, respectively.

**Network architecture.** As figure12 shows, each agent in MADDPG contain independent critic network and actor network. Our code is modified upon the public MADDPG code to fit in our environment. To fair comparison, the shape of network which are 3 layer MLP. The stage cost was defined in Eq. 14. The corresponding error functions were defined in Eq.32 and Eq.33.

**Training parameters.** To fair comparison, MADDPG use the same hyperparameter as cMARL which the learning rate of $\eta = 1 \times 10^{-2}$ was used to train both critic and actor network. The optimization target error was $\epsilon = 1 \times 10^{-3}$. The discounted factor $\gamma = 0.95$. The convergence criteria are same as cMARL.

## D.5 Run time comparison

We use Matlab for all implementation and evaluation all of our methods using our computer consisting of a AMD 5900x CPU. The simulation environment was built upon OpenSim 3.3 simulator.

|  | cMARL | dHDP | COMA | MADDPG |
|---|---|---|---|---|
| Time (seconds) | 2285 | 2405 | 2300 | 2525 |

Table D.5 shows the total training time of a single trial. In general, we observe that the training time is bottle-necked by the OpenSim forward dynamic simulation. Simulation time of each gait was significantly greater than the computing time of each weights updating ($\approx 980\%$ run time on sim, $\approx 2\%$ on weight updates)

## D.6 Optimizer

In this study, we use a customized SGD with back-propagation optimizer to train the weights in the actor and critic MLP networks. As described in Appendix D, the algorithm is based on an actor-critic structure, the direct heuristic dynamic programming (dHDP) [36].

### D.6.1 Critic network

The critic network is a three-layer MLP. It takes a state-action pair as input $x \in \mathbb{R}^{m \times 1}$ and $u \in \mathbb{R}^{n \times 1}$. The approximated cost to go value:

$$\hat{Q}_i(x, u) = W_{c2,i} \varphi \left( W_{c1,i} \left[ x^T, u^T \right]^T \right) \tag{34}$$

where $W_{c1,i} \in \mathbb{R}^{6 \times m+n}$ was the weight matrix between the input layer and the hidden layer, and $W_{c2,i} \in \mathbb{R}^{1 \times 6}$ was the weight matrix between the hidden layer and the output layer at $ith$ update. And,

$$\eta_{c1} = W_{c1,i} \left[ x^T, u^T \right]^T \tag{35}$$

$$h_{c1} = \varphi \left( \eta_{c1} \right) \tag{36}$$

$$\varphi(\eta) = \frac{1 - \exp(-\eta)}{1 + \exp(-\eta)} \tag{37}$$

where $\varphi(\cdot)$ was the tan-sigmoid activation function, and $h_{c1}$ was the hidden layer output.

The prediction error $e_c \in \mathbb{R}$ of the critic network at k steps can be written as.

$$e_{c,k} = \gamma \hat{Q}_i(x_{k+1}, \pi(x_{k+1})) - [\hat{Q}_{i+1}(x_k, u_k) - U(x_k, u_k)] \tag{38}$$

To correct the prediction error, the weight update objective was to minimize the squared prediction error $E_c$, denoted as

$$E_{c,k} = \frac{1}{2} \left( e_{c,k} \right)^2 \tag{39}$$

The weight update rule for the CNN was a gradient-based adaptation given by

$$W_{i+1} = W_i + \Delta W_i \tag{40}$$

The weight updates of the hidden layer matrix $W_{c2}$ were

$$
\begin{aligned}
\Delta W_{c2,i} &= l_c \left[ -\frac{\partial E_c}{\partial W_{c2}} \right] \\
&= l_c \left[ -\frac{\partial E_c}{\partial e_c} \frac{\partial e_c}{\partial \hat{Q}} \frac{\partial \hat{Q}}{\partial W_{c2}} \right]
\end{aligned} \tag{41}
$$

The weight updates of the input layer matrix $W_{c1}$ were

$$
\begin{aligned}
\Delta W_{c1,i} &= l_c \left[ -\frac{\partial E_c}{\partial W_{c1}} \right] \\
&= l_c \left[ -\frac{\partial E_c}{\partial e_c} \frac{\partial e_c}{\partial \hat{Q}} \frac{\partial \hat{Q}}{\partial h_{c1}} \frac{\partial h_{c1}}{\partial \eta_{c1}} \frac{\partial \eta_{c1}}{\partial W_{c1}} \right]
\end{aligned} \tag{42}
$$

where $l_c > 0$ was the learning rate of the critic network.

### D.6.2 Actor network

The actor network is a three-layer MLP. It takes a state as input $x \in \mathbb{R}^{m \times 1}$. The action output is $u \in \mathbb{R}^{n \times 1}$

$$u = \varphi \left( W_{a2} * \varphi \left( W_{a1} x \right) \right) \tag{43}$$

where $W_{a1} \in \mathbb{R}^{6 \times m}$ and $W_{a2} \in \mathbb{R}^{n \times 6}$ were the weight matrices, and $\varphi(\cdot)$ was the tan-sigmoid activation function of the hidden layer and output layer. the prediction error of actor is

$$e_{a,k} = \frac{1}{2} \left( Q_i \left( x_k, u_k \right) \right)^2 \tag{44}$$

The squared prediction error $E_a, k$, denoted as

$$E_{a,k} = \frac{1}{2} \left( e_{a,k} \right)^2 \tag{45}$$

Similarly, the weight matrix was updated based on gradient descent:

$$W_{i+1} = W_i + \Delta W_i \tag{46}$$

The weight updates of the hidden layer matrix $W_{a2,i}$ were

$$
\begin{aligned}
\Delta W_{a2,i} &= l_a \left[ -\frac{\partial E_a}{\partial W_{a2,i}} \right] \\
&= l_c \left[ -\frac{\partial E_a}{\partial e_a} \frac{\partial e_a}{\partial \hat{Q}} \frac{\partial \hat{Q}}{\partial h_{c1}} \frac{\partial h_{c1}}{\partial \eta_{c1}} \frac{\partial \eta_{c1}}{\partial u} \frac{\partial u}{\partial W_{a2,i}} \right]
\end{aligned} \tag{47}
$$

The weight updates of the input layer matrix $W_{a1,i}$ were

$$
\begin{aligned}
\Delta W_{a1,i} &= l_a \left[ -\frac{\partial E_a}{\partial W_{a1,i}} \right] \\
&= l_a \left[ -\frac{\partial E_a}{\partial e_a} \frac{\partial e_a}{\partial \hat{Q}} \frac{\partial \hat{Q}}{\partial h_{c1}} \frac{\partial h_{c1}}{\partial \eta_{c1}} \frac{\partial \eta_{c1}}{\partial u} \frac{\partial u}{\partial h_{a2}} \frac{\partial h_{a2}}{\partial \eta_{a1}} \frac{\partial \eta_{a1}}{\partial W_{a1,i}} \right]
\end{aligned} \tag{48}
$$

where $l_a > 0$ is the learning rate of the actor.

# Appendix E   Performance evaluation

## E.1   Training and evaluation

A trial consists of 300 gait cycles of continuous walking, which is chosen according to published results based on single-agent design studies using simulations and human experiments. Each gait cycle will generate one data tuple $(x_k, u_k, v_k, U_v, U_u, x_{k+1})$. A trial is considered successful if all state variables in $x_k$ reach their respective error tolerance bound (Appendix A.4, Table 2, bottom row). Additionally, one gait phase at a time, if 8 out of 10 consecutive gaits meet the state error tolerance bound condition, training has converged. If all 4 phases have converged within 300 gait cycles, the trial is a success. Results reported in this paper are based on 16 trials of OpenSim simulations for each study below.

Each of the three approaches used the same random seeds during training and evaluation. Evaluation trials are performed on 5 randomly selected ones out of the 16 training trials, and evaluations are conducted every 2 gaits during the first 50 gaits and then every 10 gaits afterwards, each using the latest policy at the time of evaluation. Due to very low success rate of the tailored MADDPG method and very low chance of retrieving a successful policy (Figure 3, grey lines), also that its current implementation may be viewed as a decomposed COMA, we have removed MADDPG from evaluation.

## E.2   Evaluation metrics

In order to ensure that amputee subjects walk safely and continuously, we consider several performance metrics: 1) This is an optimal control problem and thus the objective is to minimize regulation cost (close to 0 is better). 2) Peak angle can directly reflect amputee safety, as small peak angle difference prevents amputee stumbling in swing phase and unbalance in stance phase (close to 0 is better). 3) Symmetry in walking can prevent secondary injury to the amputee (close to 0 is better). 4) Faster learning in terms of fewer tuning steps is practically important to amputees wearing a prosthesis (fewer steps is better). 5) High success rate boosts amputees' confidence to our method and thus perform more natural walking pattern (higher is better). 6) Standard division for each metrics above are a common measure in RL (smaller is better).

1) This is an optimal control problem and thus the objective is to minimize regulation cost, which is unlike game problems to achieve maximum scores. As such, learning convergence is based on the same criteria in Appendix Table 2, We thus can compare learning and success rates by reaching the same convergence level

2) Additionally, maintaining human-prosthesis stability during walking is a must and is ensured in our design (AppendixA.4 for safety consideration) . We consider this is a qualitative measure, which is partially reflected in 2) success rate.

3) Faster learning means less tuning steps which is practically important to amputees wearing a prosthesis. They will use less effort and high success rate boosts amputees' confidence and thus natural walking patterns. Symmetry is to prevent secondary injury, while variance reduction is a common measure in RL. As a reference point, in current clinical practice, a highly skilled prosthesis needs to spend hours and multiple clinic visits to arduously hand-tune the impedance parameters for each user for a small number of walking tasks (most of which considered in our study).

Note that Energy consumption measure may not be appropriate for amputees. This measurement is too slow and may not be reliable for human-prosthesis control updates (requires  X10 minutes per sample) as it susceptible to various contamination due to several confounding factors stemming from a person's physical, physiological, and psychological condition. Additional uncertainty in using energy cost for amputee subjects can come from prosthesis fitting and related conditions.

Even though some applications such as exoskeleton control (for healthy subjects) use energy as performance goal, it is chosen rather intuitively as it is unclear whether it is influenced by human and/or robot. One important note is that metabolic cost has not been successfully demonstrated in the control of wearable robotic prostheses in walking [84].

## Appendix F    Summary of common approaches to shared autonomy

The human-prosthesis problem under our consideration falls into the area of physical human-robot interaction (pHRI) as the human and the robotic prosthesis are in direct contact at all times. It is not, however, the extensively studied pHRI problem archetype (e.g., cooperative object manipulation, human operating in a remote environment), wherein interactions are usually mediated by a third object.

For similar reasons, our pHRI problem is different in several important aspects from the scope of shared autonomy problems. We illustrate below why the problem cannot be solved by existing approaches in the literature.

1) Cross-training, such as [46] in which the cooperative tasks of placing screws and drilling screws are switched between the human and robot, cannot be applied to the human-prosthesis problem. The human and prosthesis have distinct (cooperative) roles, and there is no way to partition the walking task for role interchange.

2) Bounded memory adaptation, used in a cooperative table clearing task [47] and a moving-a-table-through-door task [85], is also not applicable. In these studies, the control was designed with the robot having specific preferences and adapting based on direct human input. In our application, the exact numerical values of the impedance parameters are not intuitive for the human to edit/update directly.

3) The predict and blend approach [86, 48] to enable robot autonomy and user input or blend the two is also problematic in this setting. In these studies, the human has the lead role (e.g. specifying what pose to catch or which object to catch), and robot's role is to facilitate completing the task. In locomotion, there is no clear single goal from the perspective of the human, and it is difficult to furnish user inputs which substantively inform the controller to improve gait synchronization (cf. Point P4 of Common Issues).

4) Lastly, model-free deep RL [49], where shared autonomy is achieved by training an end-to-end mapping from states and user inputs to agent action choices, is not conformable to our problem either. Fundamentally, in these methods the robot's role is reactive to the states/controls generated by the human, and attention is not paid to the human learning and its intrinsic coupling to the robot learning. In locomotion, human and robot are equal partners and learn simultaneously.

As shown, our pHRI control problem does not fit into those extensively studied frameworks. Additionally, the first 3 approaches require a known dynamic model, a set of possible goals, and a known user policy under a specific goal. These are too restrictive or unrealistic for human-prosthesis problem. Modeling a human-robotic prosthesis system is challenging if not at all possible. Neither do we know how human and a robotic leg interact. Even though the 4th approach does not require a model as it rely on big data, it is still not feasible for amputee subjects as big data for individual subjects is not available, and scaling from 12 control to hundreds if we are to design controls of knee, ankle and hip joints would be unscalable.

## Appendix G    Publicly Available Code

Code for creating the human-robot environment and the original dHDP actor-critic learning algorithm are provided at https://github.com/JennieSi-Lab-RLOC/NeurIPS2022/tree/main/OpenSim%20Model

The following references go along with Openreview discussion:

[15] Minhan Li, Yue Wen, Xiang Gao, Jennie Si, and He Helen Huang. Towards expedited impedance tuning of a robotic prosthesis for personalized gait assistance by reinforcement learning control. arXiv preprint arXiv:2006.06518, 2020.

[16] Yue Wen, Minhan Li, Jennie Si, and He Huang. Wearer-prosthesis interaction for symmetrical gait: a study enabled by reinforcement learning prosthesis control. IEEE transactions on neural systems and rehabilitation engineering, 28(4):904–913, 2020.

[37] Yue Wen, Jennie Si, Xiang Gao, Stephanie Huang, and He Helen Huang. A new powered lower limb prosthesis control framework based on adaptive dynamic programming. IEEE transactions on neural networks and learning systems, 28(9):2215–2220, 2016.

[38] Yue Wen, Jennie Si, Andrea Brandt, Xiang Gao, and He Huang. Online reinforcement learning control for the personalization of a robotic knee prosthesis. IEEE transactions on cybernetics, 2019.

[39] Wentao Liu, Junmin Zhong, Ruofan Wu, Bretta L Fylstra, Jennie Si, and He Helen Huang. Inferring human-robot performance objectives during locomotion using inverse reinforcement learning and inverse optimal control. IEEE Robotics and Automation Letters, 2022.

[40] Yue Wen, Andrea Brandt, Ming Liu, He Huang, and Jennie Si. Comparing parallel and sequential control parameter tuning for a powered knee prosthesis. In 2017 IEEE International Conference on Systems, Man, and Cybernetics (SMC), pages 1716–1721. IEEE, 2017.

[41] Yue Wen, Andrea Brandt, Jennie Si, and He Helen Huang. Automatically customizing a powered knee prosthesis with human in the loop using adaptive dynamic programming. In 2017 International Symposium on Wearable Robotics and Rehabilitation (WeRob), pages 1–2. IEEE, 2017.

[42] Xiang Gao, Jennie Si, Yue Wen, Minhan Li, and He Huang. Reinforcement learning control of robotic knee with human-in-the-loop by flexible policy iteration. IEEE Transactions on Neural Networks and Learning Systems, 2021.

[43] Minhan Li, Xiang Gao, Yue Wen, Jennie Si, and He Helen Huang. Offline policy iteration based reinforcement learning controller for online robotic knee prosthesis parameter tuning. In 2019 International Conference on Robotics and Automation (ICRA), pages 2831–2837. IEEE, 2019.

[44] Yue Wen, Xiang Gao, Jennie Si, Andrea Brandt, Minhan Li, and He Helen Huang. Robotic knee prosthesis real-time control using reinforcement learning with human in the loop. In In- ternational Conference on Cognitive Systems and Signal Processing, pages 463–473. Springer, 2018.

[45] Yue Wen, Ming Liu, Jennie Si, and He Helen Huang. Adaptive control of powered transfemoral prostheses based on adaptive dynamic programming. In 2016 38th Annual International Conference of the IEEE Engineering in Medicine and Biology Society (EMBC), pages 5071– 5074. IEEE, 2016.

[46] Ruofan Wu, Minhan Li, Zhikai Yao, Jennie Si, et al. Reinforcement learning enabled automatic impedance control of a robotic knee prosthesis to mimic the intact knee motion in a co-adapting environment. arXiv preprint arXiv:2101.03487, 2021

[60] Hogan, Neville. "Impedance control: An approach to manipulation." 1984 American control conference. IEEE, 1984.

[62] Taga, Gentaro, Yoko Yamaguchi, and Hiroshi Shimizu. "Self-organized control of bipedal locomotion by neural oscillators in unpredictable environment." Biological cybernetics 65.3 (1991): 147-159.

[71] M Jason Highsmith, Casey R Andrews, Claire Millman, Ashley Fuller, Jason T Kahle, Tyler D Klenow, Katherine L Lewis, Rachel C Bradley, and John J Orriola. Gait training interventions for lower extremity amputees: a systematic literature review. Technology & Innovation, 18(2-3):99–113, 2016.

[72] Alberto Esquenazi. Gait analysis in lower-limb amputation and prosthetic rehabilitation. Physical Medicine and Rehabilitation Clinics, 25(1):153–167, 2014.

[73] George NS Marinakis. Interlimb symmetry of traumatic unilateral transtibial amputees wearing two different prosthetic feet in the early rehabilitation stage. Journal of rehabilitation research and development, 41(4):581–590, 2004.

[79] Geyer, Hartmut, Andre Seyfarth, and Reinhard Blickhan. "Positive force feedback in bouncing gaits?." Proceedings of the Royal Society of London. Series B: Biological Sciences 270.1529 (2003): 2173-2183.

[80] Shamaei, Kamran, Gregory S. Sawicki, and Aaron M. Dollar. "Estimation of quasi-stiffness of the human knee in the stance phase of walking." PloS one 8.3 (2013): e59993.

[81] Minhan Li et al. Learning based control adaptation of robotic knee prostheses. 2022

[86] Wentao Liu, Ruofan Wu, Jennie Si, and He Huang. A new robotic knee impedance control parameter optimization method facilitated by inverse reinforcement learning. IEEE Robotics and Automation Letters, pages 1–8, 2022.

[89] Zhang, Juanjuan, et al. "Human-in-the-loop optimization of exoskeleton assistance during walking." Science 356.6344 (2017): 1280-1284.

[105] Cara Gonzalez Welker, Alexandra S Voloshina, Vincent L Chiu, and Steven H Collins. Short-comings of human-in-the-loop optimization of an ankle-foot prosthesis emulator: a case series. Royal Society open science, 8(5):202020, 2021.

[106] Tyler R Clites, Max K Shepherd, Kimberly A Ingraham, Leslie Wontorcik, and Elliott J Rouse. Understanding patient preference in prosthetic ankle stiffness. Journal of neuroengineering and rehabilitation, 18(1):1–16, 2021.

[107]Abbas Alili, Varun Nalam, Minhan Li, Ming Liu, Jennie Si, and He Helen Huang. User controlled interface for tuning robotic knee prosthesis. In 2021 IEEE/RSJ International Conference on Intelligent Robots and Systems (IROS), pages 6190–6195. IEEE, 2021.

[108] Kejun Li, Maegan Tucker, Erdem Bıyık, Ellen Novoseller, Joel W Burdick, Yanan Sui, Dorsa Sadigh, Yisong Yue, and Aaron D Ames. Roial: Region of interest active learning for characterizing exoskeleton gait preference landscapes. In 2021 IEEE International Conference on Robotics and Automation (ICRA), pages 3212–3218. IEEE, 2021.

[109] Maegan Tucker, Ellen Novoseller, Claudia Kann, Yanan Sui, Yisong Yue, Joel W Burdick, and Aaron D Ames. Preference-based learning for exoskeleton gait optimization. In 2020 IEEE international conference on robotics and automation (ICRA), pages 2351–2357. IEEE, 2020