# OpenReview forum: "Human-Robotic Prosthesis as Collaborating Agents for Symmetrical Walking"
_NeurIPS.cc/2022/Conference — NeurIPS 2022 Accept_

### Official Review · Reviewer_Wv4V · 2022-07-10

**Rating:** 7
**Confidence:** 4
**Soundness:** 3 good
**Presentation:** 3 good
**Contribution:** 3 good

**Summary:**

The paper presents an application for human-robot collaboration and shared autonomy — the task of robotic prosthesis for symmetrical walking — which has not been studied extensively using RL. The authors propose a collaborative MARL framework to study the problem by defining a state-space problem and joint cost function, and show that an actor-critic policy can jointly optimize this problem. Compared to some relevant baselines like COMA and MADDPG, the proposed agent performs better in terms of learning a stable gait. Useful ablation analysis justifies some of the design decisions made.


**Questions:**

Drawing from the discussion of weaknesses above, here are some clarifications/suggestions that would help me re-evaluate the paper after authors’ response. I am more than willing to engage with the authors and modify my review score contingent on the author discussion phase.
1. Improving interpretability of results, with a better metric or a qualitative visualization/analysis.
2. Adding implementation details, or ideally, releasing the code with the paper.
3. A discussion on positioning wrt shared autonomy; ideally, added comparison against some existing approach(es).
4. [Line 355] What are some other, more natural, objectives that may be used for providing shared goals in the proposed cMARL framework for robot prosthesis? Beyond the task of prosthesis, what broader set of collaborative objectives in HRI can the proposed method be applied for? A discussion on this would greatly improve how the broader community receives and builds on this work.

--------

After a productive round of responses from the authors, I have updated my score to recommend acceptance.

**Limitations:**



- It does not seem that the implementation/code is shared in the supplemental material. The data being proprietary is understandable, but that is not a sufficient reason to not release the code. The paper is also sparse in implementation details of cMARL, as well as the baselines, and it may be really difficult for people in sibling disciplines of HRI to draw useful insights and quickly try out the proposed algorithm.
- I could not find an explicit discussions on societal impacts of the work. While the immediate impacts of the work are not particularly troubling, I believe any paper on HRI, esp. involving prosthesis, should include a discussion on the potential negative implications and a discussion of how the proposed algorithms may be susceptible to malicious attacks. While this is not a criticism/limitation of the research proposed (and does not factor into my review), it helps people outside the community contextualize the research and its potential limitations.


**Strengths And Weaknesses:**

_Heads up_: I am not familiar with the space of human-robot prosthesis, and my review is from the perspective of a general robotics/HRI/ML researcher. A quick skim of the space suggests some prior work in using RL for limb/knee prosthesis, but the proposed collaborative formulation seems novel to me.

Strengths
---
- The paper does a great job of describing the problem of human-robot prosthesis to an outsider (me), and it was possible to follow along the terminology and problem formulation with some effort. It’s always hard to write an accessible paper bridging communities and this paper does a good job of that.
- The experiments seem very well designed and answer specific questions I have as a reader. The ablation and reliability studies were particularly useful in convincing me that (i) the cost function is doing what it should, (ii) the human input is actually being used in a non-trivial manner, and (iii) the method can adapt to more general target signals (velocity, slope etc.). Learning curves alone do not capture this story, and the analysis included by the authors seems really comprehensive and useful.
***
Weaknesses
---
- [Interpreting Results] From the quantitative results in the paper, it is very hard to interpret what is a “good enough” error/symmetry. Is the lowest always better, or is there a point below which it does not matter and should be treated as the “target”. It would really help to include some qualitative result denoting what it means to “do better” in the task, e.g. stability of learned gaits, or energy consumption estimate etc. Also, the learning curves seem to only denote the training curves and not the validation curves — can the authors provide validation curves instead/alongside to convey the full picture? Training loss going down does not capture the entire process.
- [Implementation Details] I found the paper severely lacking in implementation details, and in its current shape, the paper is not reproducible. Beyond providing the form of the agent and objective, the paper has no information on the agent architectures used, training algorithm, optimizer, amount of training time etc.
- [Related Work: Shared Autonomy] The discussion on related work seems to miss a very important body of research in shared autonomy [1, 2, 3, to name a few…]. The task of shared autonomy very closely resembles the problem setup in the paper — human input combined with a (semi-)autonomous agent to achieve a common goal — and should be discussed when positioning the paper. While the proposed approach is shown to be better than other MARL baselines, it would also benefit from a comparison with a more shared autonomy-esque viewpoint to the problem — seeing an empirical evaluation against this, or a discussion of why this is not feasible, would really strengthen the paper.
***
References
---
[1] Nikolaidis and Shah, “Human-Robot Cross-Training: Computational Formulation, Modeling and Evaluation of a Human Team Training Strategy” (2013)

[2] Javdani, Srinivasa, and Bagnell, “Shared Autonomy via Hindsight Optimization” (2015)

[3] Reddy, Dragan, and Levine, “Shared Autonomy via Deep Reinforcement Learning” (2018)

---

> ### Author Response · Authors · 2022-08-02
> **We thank the reviewer for recognizing novelty in this work and for questions related to performance, cMARL algorithm, implementation, and our contribution in the context of shared autonomy**
>
> >P3.1 From the quantitative results in the paper, it is very hard to interpret what is a “good enough” error/symmetry. Is the lowest always better, or is there a point below which it does not matter and should be treated as the “target”. It would really help to include some qualitative result denoting what it means to “do better” in the task, e.g. stability of learned gaits, or energy consumption estimate etc. Also, the learning curves seem to only denote the training curves and not the validation curves — can the authors provide validation curves instead/alongside to convey the full picture? Training loss going down does not capture the entire process.
>
> >Question 1 Improving interpretability of results, with a better metric or a qualitative visualization/analysis.
>
> 1) To improve interpretability of results, we have added performance metrics (“Performance Criteria” in paper), rationale (Appendix E), and summarized the results of Table 1 in the paper to address this important question.
>
> 2) We refer the reviewer to general point G4.3 and G4.4 for detailed discussions of the energy consumption and balance confidence metrics, respectively.
>
> 3) On Validation Curves. Please refer to Figure 5. Validations were conducted every 2 gaits during the first 50 gaits and then every 10 gaits afterwards. The y-axis includes total cost of the validation trials.
>
>
> >P3.2 Question 2.	Adding implementation details, or ideally, releasing the code with the paper.
>
> Please see general point G1 for a detailed implementation description on this important issue.
>
> >P3.3 Question 3. A discussion on positioning wrt shared autonomy; ideally, added comparison against some existing approach(es).
>
> Please refer to “Related Work” on “Shared Autonomy” in the paper. Also please refer to Appendix F where we provide a detailed analysis of why current treatise of pHRI does not apply to our HPC problem, and thus why we need an innovative solution as in this study.
>
> >P3.4 Question 4. What are some other, more natural, objectives that may be used for providing shared goals in the proposed cMARL framework for robot prostheses? Beyond the task of the prosthesis, what broader set of collaborative objectives in HRI can the proposed method be applied for? A discussion on this would greatly improve how the broader community receives and builds on this work.
>
> 1) Please refer to General point G4 for examples of other complex human objectives, and why amputee subjects prioritize differently in varying scenarios/environments, and thus show why our formulated cost involves kinematics of walking, symmetry, and direct human influence in this work. Please refer to response P3.1 above.
>
> 2) Our method can substantively address the applications we mention in the Shared Autonomy section. For example, the placing screws and drilling task, cooperative table cleaning task, and moving table through door task.
>
> 3) More broadly speaking, if a given HRI application can clearly define their human and robot state-control variables as well as a reasonable collective objective function, our algorithm can be applied. However, for applications we mentioned above, human trust will be the challenge because for our cMARL formulation, human prostheses rely on each other to walk and trust will not be the consideration. Most recent existing work [92] shows that trust may be dependent mainly on early interactions with the system, while the agent is still developing competency. Without trust, it is relatively hard to develop a successful model. As has been said, in order to solve these tasks, the objectives in our formulation need to be expanded and human trust needs to be formally considered.
>
> >P3.5 will be discussed in following comment window:

---

> > ### Author Response · Authors · 2022-08-03
> > **Following Reviewer Point P3.5:**
> >
> > > P3.5 I could not find an explicit discussion on the societal impacts of the work. While the immediate impacts of the work are not particularly troubling, I believe any paper on HRI, esp. involving prosthesis, should include a discussion on the potential negative implications and a discussion of how the proposed algorithms may be susceptible to malicious attacks. While this is not a criticism/limitation of the research proposed (and does not factor into my review), it helps people outside the community contextualize the research and its potential limitations.
> >
> > This research has direct and positive societal impact. In the US, approximately 1.7 million people live with limb loss [90] with lower limb loss as a majority. The amputee population is expected to double by 2050 as the population ages and the incidence of dysvascular disease increases [90]. As most lower limb amputees use prosthetic legs to restore basic bipedal locomotion [91], our solution to the prosthesis control problem can potentially help improve the function and quality of life of lower limb amputees.
> >
> > Potential Negative Impacts: Potential negative impact such as a malicious attack may not be of concern as we have several important measures.
> >
> > 1) Our controller tuning can take place on an embedded local computer for on-board learning.
> >
> > 2) Our extensive safety/reliability measures as described in Appendix A.4 and response P1. Additionally, it is conceivable to dedicate a control button (can be operated by amputee) to return to default control and even cut off from local computer.

---

> > ### Comment · Reviewer_Wv4V · 2022-08-05
> > **Response to authors**
> >
> > - [Interpretation of Results] This discussion is super useful, especially for someone unfamiliar with the problem constraints. The response addresses my concerns here and I *strongly recommend* including some form of this discussion (esp. G4.3 and G4.4) in either the main paper or an appendix.
> >
> > - [Implementation Details] The changes address my concerns here. I would encourage the authors to release the code (esp. on acceptance), [as stated in the NeurIPS 2022 guidelines](https://neurips.cc/Conferences/2022/PaperInformation/CodeSubmissionPolicy).
> >
> > - [Related Work] I have no further concerns here.
> >
> > - [Other Objectives] This is a great discussion, I would strongly encourage the authors to include this in the main paper.
> >
> > - [Societal Impacts] This is a great discussion, I would strongly encourage the authors to include this in the main paper (even briefly).
> >
> >
> > Thank you for the very detailed (and useful) responses to my questions and concerns. I am very happy to recommend acceptance of the paper and have updated my score accordingly.

---

> > > ### Author Response · Authors · 2022-08-05
> > > **Thank you so much for all your valuable comments and feedback. All points are well taken.**
> > >
> > > [Interpretation of Results] -- great that we were able to be "super helpful". Your questions were super helpful to us as well. we will do our best to include materials esp. G4.3 and G4.4.
> > > [Implementation Details] -- we already agreed to that.
> > > [Related Work] -- thank you.
> > > [Other Objectives] and [Societal Impacts] -- we will do our best here.
> > > It has been a great experience to have this intellectual exchange with you.

---

### Official Review · Reviewer_bfd4 · 2022-07-10

**Rating:** 5
**Confidence:** 4
**Soundness:** 2 fair
**Presentation:** 3 good
**Contribution:** 2 fair

**Summary:**

The paper studies the problem of controlling robotic lower limb prosthesis for achieving better human walking assistance. The main idea is a formulation of multi-agent RL, where one agent represents the human controller and the other represents the robotic prosthesis. The robotic controller has access to some information about the human state and controller, which helps it better assist the human walking. To optimize the controllers, a Q-learning-based algorithm is deployed. The proposed method is evaluated in the OpenSim simulation environment and compared to alternative multi-agent RL algorithms. An ablation study is also carried out to investigate the impact of including human controller action as observation for the robotic controller.

**Questions:**

- During the walking, who decides the desired walking speed? If human determines that, how does the robot have access to it?
- How well does the method generalize to unseen human behavior? For example, if the thresholds in the human controller state machine is varied during testing.
- In the ablation it appears that without human controller input the model learns slower, but converges to a similar level of performance. Some discussion on this would be useful to better understand the impact of including the additional information.
- Human users can adapt their behavior to the prosthesis device. To account for this, one need to either model the adaptation process, or obtain a model that is robust to this. How would the proposed method handle that?
- How does the proposed method compare to a biomechanical engineering-based approach in terms of performance?

**Limitations:**

The current discussion on limitations of the method is good in general.

**Strengths And Weaknesses:**

Strengths:
- The paper studies an important and practical problem of controller robotic prosthesis for human motion assistance.
- The idea of multi-agent RL training for the robotic controller seems reasonable.
- The proposed formulation obtained promising learning performance in a simulated task.

Weaknesses:
- The technical contribution of the paper is not very clear. The main innovation seems to be the formulation of the observation and action spaces for the human and robotic agents and how the robot agent can observe certain information about the human. However, without evaluation on real humans, it’s hard to tell how realistic these assumptions are.
- The proposed formulation seems specific to the setup: if the task becomes walking up/down stairs, or traversing stepping stones, having desired and commanded velocity from the human controller might be insufficient.

---

> ### Author Response · Authors · 2022-08-02
> **We greatly appreciate the reviewer’s questions that have helped us highlight the significance of the research and our contribution**
>
> > P 2.1: The technical contribution of the paper is not very clear. The main innovation seems to be the formulation of the observation and action spaces. However, without evaluation on real humans, it’s hard to tell how realistic these assumptions are.
>
> 1) Please refer to our Contributions section in the revised paper for this important question.
>
> 2)  As we pointed out in “Shared Autonomy” under “Related Study and Challenges”, our HPC problem is uniquely challenging and different from the current (p)HRI problems that have been the focus in many studies. As such, pHRI problems such as upper limb wearables and lower limb exoskeletons are fundamentally different problems. As such, this is a new dimension to be researched in pHRI and thus this work innovative in the way as summarized in “Contributions”.
>
>
>
> > P2.2 The proposed formulation seems specific to the setup: if the task becomes walking up/down stairs, or traversing stepping stones, having desired and commanded velocity from the human controller might be insufficient.
>
> 1)  We believe there may be a misunderstanding here. Our control design does not require "desired and commanded velocity from human controller". Note that the "desired" gait trajectory or velocity is up to the human as shown in the states of the robot control system. The robot joint motion is to follow the human's. The reviewer may have mixed up how simulations were setup vs a real human in experiment. In OpenSim, we have to set up a controller for human intact knee as the default setup only offers a normative knee joint movement, a case that does not reflect realistic human-prosthesis walking. This same setup has been reported in previous studies using OpenSim simulations [70] and real human experiments [85].
>
> 2)  If we change terrain or task significantly, which includes traversing stepping stones, we will need a "task planner", which is not the focus of this study. Please also refer to P1.5.
>
> > P2.3 Question 1: During the walking, who decides the desired walking speed? If human determines that, how does the robot have access to it?
>
> A subject (simulated or real) walks at their preferred speed, and thus speed is determined by the human. The load cells in the ankle provides readings of ground reaction force (GRF) which is used to determine stance time and full gait duration. The step length is captured by Vicon motion system from which we measure speed information.
>
> >P2.4 Question 2: How well does the method generalize to unseen human behavior? For example, if the thresholds in the human controller state machine is varied during testing.
>
> 1) Please refer to response P1.4 for issue on "unseen" human behavior. In essence, three measures are in place to ensure human safety in major "unseen" human behavior such as bumping into an obstacle.
>
> 2) On the example scenario of varying thresholds in the human FSM, given a task, the thresholds don’t need to change. The FSM-IC framework was designed to restore normative gait. The switching rules between finite states must be well defined and measurable [63]. These rules are derived from body-mass-normalized data [86] and therefore don't change by person.
>
> > P2.5 Question 3: In the ablation it appears that without human controller input the model learns slower, but converges to a similar level of performance. Some discussion on this would be useful to better understand the impact of including the additional information.
>
> 1)  This is an optimal control problem and thus the objective is to minimize regulation cost, which is unlike game problems to achieve maximum scores. As such, learning convergence is based on the same criteria in Table 2, Appendix A.4. We thus can compare learning and success rates by reaching the same convergence level.
>
> 2) Please see general point G3 for a detailed description and interpretation of each of our learning performance metrics, including learning rate.
>
> > P2.6 Question 4: Human users can adapt their behavior to the prosthesis device. To account for this, one need to either model the adaptation process, or obtain a model that is robust to this. How would the proposed method handle that?
>
> 1) In a nutshell, FSM-IC integrated with cMARL are important ingredients of how we handled this complex co-adapting human-robot control problem. Our general responses G1~G4 together, may shed some further insight on this important question.
>
> 2) Under “Related Work and Challenges”, we added “Shared Autonomy”, “Modeling Challenges” and “Utility Challenges. Together, we showed why our HPC problem is uniquely challenging, and thus it shows our proposed approach is innovative.
>
> >P2.7 Question 5 will be discussed in following comment window:

---

> > ### Author Response · Authors · 2022-08-03
> > **Following Reviewer Point 2.7 Question 5:**
> >
> > > P2.7 Question 5: How does the proposed method compare to a biomechanical engineering-based approach in terms of performance?
> >
> > Successful demonstrations of continuous and stable walking with robotic prosthesis are rare prior to some recent results centered on RL based control approaches. The concept of virtual constraint (VC) may lead to a solution to the prosthesis control problem. It generates coordinated joint motions as target joint profiles including robotic knee [87]. The idea was to describe the geometric relationships among joints which could be encoded by hybrid zero dynamics [88]. This idea could be used in our current framework to replace the robotic knee target profile. On the other hand, it may introduce a new issue as the VC model was built on data from healthy subjects. Published reviews of other biomechanical engineering-based approaches can be found in Introduction of [38,46]

---

### Official Review · Reviewer_FF9f · 2022-07-13

**Rating:** 6
**Confidence:** 4
**Soundness:** 3 good
**Presentation:** 3 good
**Contribution:** 3 good

**Summary:**

This paper presents an approach to adapting control parameters in a prosthetic limb using a multi-agent Reinforcement Learning design that considers the human and the prosthetic as collaborating agents with the goal of achieving a symmetric gait pattern.
The design is evaluated and compared in a number of simulation experiments.

**Questions:**

Some questions arise regarding the degree to which the evaluations are realistic:
To what degree is the simulation model for the human realistic ?
To what degree do the learned impedance parameters generalize to slight variations of the terrain ?
Is it realistic to assume that the human has the same objective to achieve symmetric patterns ?

To what degree can the approach be generalized to other non-straight walking tasks?

**Limitations:**

The paper addresses social impacts and considerations and discusses limitations in the conclusions. However, it would have been nice to see more discussion regarding the limitations (and potential ways to address them) earlier in the paper, especially in the experiments section to allow the reader a clearer picture regarding what the results show (and what they don’t capture).

**Strengths And Weaknesses:**

The control of prosthetics in a way that is natural and does not put increased burden on the human is can have significant impact on the
life is amputees. Moreover, learning collaborative control tasks with humans in general is an important field. The paper is generally well written and lays out the rationale for the approach well.
The main weakness of the paper is in the experimental evaluation which, as it is done in simulation with a ver simple (and beneficial to the approach) human model. In particular it assumes that the human learns and adapts in the same way as the prosthetic (using the partial gradient of the prescribed control model with respect to the joined utility). This seems to remove some of the rationale of the interactive multi-agent approach that is to react to and address the human’s reactions to the behavior of the prosthetic, which, in turn, is likely based in past experiences and existing control schemes in the human. A discussion as to the assumptions in the simulation (especially with respect to the human model) both in terms of the utility used - don’t humans likely have more complex utilities?- and the policy- a human would start with a strongly pre-trained system with expectations about the other limb - or , better, inclusion of experiments with pre-trained human models that reflect likely human reactions to “un-natural” controls especially early on (and an evaluation of the difference this makes on learning speed and performance) would be very useful.
Also, it would be useful if the authors would include a comparison of the learned impedance parameter functions for the 3 scenarios and the base performance of the flat ground parameters in the robustness scenarios to give the reader an idea whether the approach would have to be trained for all terrain types separately (and maybe ultimately require a “terrain classifier”) or whether general parameters for variable terrains can be learned.
Also, to better understand the generality of the approach across walking scenarios, a discussion of whether and how it could be adapted to settings (like turning) where symmetry is not the main criterion would be useful.

---

> ### Author Response · Authors · 2022-08-02
> **We greatly appreciate the reviewer's questions centered on simulation model, human aspects, and task learning during cMARL**
>
> > P1.1: Question 1: To what degree is the simulation model for the human realistic ?
>
> Please refer to our response to this important issue in G2 above
>
> > P1.2 it assumes that the human learns and adapts in the same way as the prosthetic (using the partial gradient of the prescribed control model with respect to the joined utility). This seems to remove some of the rationale of the interactive multi-agent approach that is to react to and address the human’s reactions to the behavior of the prosthetic
>
> Please refer to the cMARL structure in Fig. 1. The human policy differs from robot policy. Also, see explanations from line 175 to line 207 in the paper where we explain human policy/control. Lastly, please see general point G4 for a detailed discussion on human objective and utility related to the proposed HPC problem solution. In essence, we model the human and robot as independent yet collaborating agents working toward the same goal. In reality, modeling amputee-prosthesis interaction is nearly impossible as little is known. We therefore do not have the luxury as in the vast literature on shared autonomy where performance goal is well defined according to task end point (please refer to “Related Work and Challenges”). This motivates us to develop this innovative approach to address the HPC problem.
>
> > P1.3: don’t humans likely have more complex utilities?
>
> Please see general point G4.2 and Modeling and Utility Challenges in the paper about how we formulate the cost and treat complex human utilities for our HPC problem.
>
> > P1.4 a human would start with a strongly pre-trained system with expectations about the other limb - or , better, inclusion of experiments with pre-trained human models that reflect likely human reactions to “un-natural” controls especially early on (and an evaluation of the difference this makes on learning speed and performance) would be very useful
>
> 1) It is common in a benchmark environment such as humanoid in OpenAI gym that "un-natural" controls are accounted for in initial training. However, safety must be ensured for the subject in human tests at all times (cf. Appendix A.4 for safety considerations). In our FSM-IC framework, "un-natural" controls are avoided by the 3 levels of safety guarantees (Appendix A.4).
>
> 2) Regarding pre-trained systems: please refer to the introduction for challenges of modeling amputee-prosthesis behavior which provides a context of why we develop this innovative solution approach.
>
> > P1.5 Comparison of the learned impedance parameter functions for the 3 scenarios. Question 2: To what degree do the learned impedance parameters generalize to slight variations of the terrain?
>
> We have included Table 3 in Appendix B.3 to summarize the learned impedance parameters for the 3 scenarios. On variations of terrain.
> 1)  If terrain/task change is significant (e.g., from level ground to stairs or turning), we will need an additional module of task planning as human joint movement profiles change significantly and thus controllers are expected to be different.
>
> 2)  Existing OpenSim and human testing show that learned impedance control do generalize to slight variations such as level to small slopes or pace change [44, 46, 86].
>
> > P1.6: Question 3. Is it realistic to assume that the human has the same objective to achieve symmetric patterns?
>
> Symmetry is necessary. Asymmetrical gait is frequently reported in people with unilateral lower limb amputation [ 81 , 82], and is associated with many secondary issues, such as osteoarthritis of unamputated joints [83] and lower back pain [84].
>
> > P1.7 it would have been nice to see more discussion regarding the limitations (and potential ways to address them) earlier in the paper, especially in the experiments section to allow the reader a clearer picture regarding what the results show (and what they don’t capture).
>
> We have included "Limitations of current results" at the end of the "Experiment" section as suggested.

---

### Author Response · Authors · 2022-08-02
**We thank the editors and reviewers for insightful comments which have helped us greatly to clarify our contributions. We have substantively revised the paper to reflect these comments. We greatly appreciate the opportunity of rebuttal and we look forward to further discussions if needed. Below is a highlight of our response to some common issues**

We refer the reviewers especially to the Introduction, Related work, and Contributions sections for significant revisions. We added Performance Metrics and Summary Evaluation Results among other clarifications. Also, we added discussion of the Modeling Challenges and Human Utility Challenges which make our HPC problem inherently difficult. Finally, we refer the reviewers to the Shared Autonomy section, in which we distinguish the new challenges present in the HPC problem which are not present in and are not addressed by existing classes of problems. Thus, our HPC framework represents a significant and novel contribution to the human/robot community.


> G1 Not enough implementation detail

We have in Appendix C and D included a detailed description of all implementation steps, including OpenSim human simulation with realistic noise data, and the cMARL control algorithm and optimizer. The section also includes OpenSim installation, human model and environment settings, code of control function of how to generate knee torque with FSM, and custom Matlab/OpenSim interface script. Code of our dHDP based controller will be released to a github repository at the time of the camera-ready version.

> G2 Is simulation model for human realistic

1) FSM impedance control (FSM-IC) captures the essential inherent properties of a mechanical joint and the underlying dynamical response characteristics of a human as they learn the features of the prosthetic in locomotion. Humans control muscle activity to adjust joint impedances in walking [60,62]. Such compliant behavior in lower limbs is fundamental to normative and safe human locomotion [79,80]. FSM-IC mimics this fundamental biological dynamic property that governs the human joint-torque relationship. Simulation models based on this physical principle ensures realistic locomotion patterns.
2)  we added real sensor noise to the simulated human/robot states $x_k$ and actuator noise to the human/robot controls $v_k$, $u_k$. They are added as white Gaussian noise with the standard deviation obtained from data collected during human experiment testing of level-ground walking. For a detailed explanation of the data collection process for noise generation, see Appendix A.
3) We agree that simulations usually do not completely render performance of a real human-robot system (cf. general point G4 below for discussion). However, simulations are critically important in addressing our HPC problem. Exploration of problem formulation, control algorithm design, and systematic evaluation take long time and are necessary to be performed prior to human experiment due to factors, such as human fatigue, human safety, human lost interest or confidence caused by repeated trial-and-error and thus lack of motivation to participate, time spent, and significant cost associated with testing amputee subjects
4) Existing experimental studies [15,16,37-47] on RL control of the robotic knee all commonly begin with OpenSim simulation to conceptualize and validate new solution approaches. After simulation validation, their controller structures are applied in human experiments. Some hyper parameters in the algorithm would not directly transfer to human experiments since multiple engineering implementation conditions need to be taken into account including, e.g. gait phase segmentation, design of convergence criteria in consideration of measurement noise and more. This study follows the same developmental steps, namely from systematic simulation to validation through real human experiment as successful demonstrations in previous existing work.

>G3 Performance measure and evaluation metric

For performance metric and rationale, please refer to "Performance Criteria" section on page 7 of the paper where we also added Table 1 to summarize performance evaluations. Please also refer to Appendix E for more detailed information.

>G4 Novelty of HPC problem and proposed framework, inherent challenges in defining human objective and utility

There are several important aspects to consider as follows (In below new comment window):

---

> ### Author Response · Authors · 2022-08-02
> **Follow common point comments G4:**
>
> > G4 Novelty of HPC problem and proposed framework, inherent challenges in defining human objective and utility
>
> 1) Under “Shared Autonomy” of “Related Work and Challenges”, we highlight the fundamental difference of our HPC problem from existing pHRI problems, a direct consequence of which is that, while it is clear what the objective or utility may be for “end point” type pHRI problems, defining human objective and utility for control purposes is a challenge. We summarized the challenges under “Related Work and Challenges”, the 2 new sections on “Modeling Challenges” and “Human Utility Challenges”.
>
> 2) In this study, we formulate human-prosthesis collaborative goal based on latest knowledge on human-prosthesis system [39,46,81]. A multitude of utilities for walking are possibly used by humans, such as energetics, walking pace, walking pattern, and more. However, for amputee subjects, being able to walk continuously (have near normative knee pattern represented by knee joint kinematics) is the first priority, and that is reflected in the knee kinematics in our current cost. Symmetry is also of paramount importance to avoid secondary injury (refer to P1.6 for details). Considering human influence in the cost is one of our contributions here.
>
> 3) On Energy Consumption. Energy consumption measures may not be appropriate for amputees. This measurement is too slow and may not be reliable for human-prosthesis control updates (requires ~X10 minutes per sample) as it is susceptible to contamination due to several confounding factors stemming from a person's physical, physiological, and psychological condition. Perhaps for these reasons, we note that metabolic cost has not been successfully demonstrated in the control of wearable robotic prostheses in walking [89].
>
> 4) Measures such as balance confidence level and walking stability have been established for gait performance evaluations in rehab clinics. But improving gait symmetry is not only a common goal for amputee rehabilitation [72,73,74], more importantly, previous study showed that symmetry related measures can be significantly influenced by impedance control [16].
>
> 5)  Studies have considered various performance measures as the goal to personalize the wearable robot control. Metabolic cost has been reported as a performance objective for optimization in exoskeleton control to improve walking energetic efficiency of non-disabled individuals [105], [91]. The resulting customized and optimized control allowed humans to eventually reduce overall energy expenditure in walking. However, limited success has been reported in using metabolic cost as the objective for optimizing robotic lower limb prosthesis control [106]. Some research groups have attempted to bring user preference into prosthetic device control by allowing users to self-select their preferred control parameters [107],[108]. Another research group proposed learning algorithms [109],[110] to include human feedback, such as good vs bad and preferred action 1 vs action 2, to identify optimal regions of an objective function in the exoskeleton control optimization process. One of the difficulties in incorporating user perception into the control objective is how to reliably quantify and mathematically describe this goal. Additionally, these optimization goals are based on human measurements only, which are chosen by intuition, not chosen systematically. Furthermore, it is not clear how to arbitrate between robotic prosthesis control and human control in a collective control goal during walking, and whether there are other confounding factors influencing the human-robot collective performance.
>
> > Note: Reference numbers are according to revised paper reference

---

### Meta-Review · Area_Chair_BDZT · 2022-08-30

**Recommendation:** Accept
**Confidence:** Less certain

**Metareview:**

This paper present an approach for control of prosthesis under a novel collaborative multi-agent formulation. The approach is demonstrated in simulation.

All the reviewers agree that the paper contribution is novel and interesting.
The reviewers also provided several suggestions on how to improve the manuscript and pointed out the limitations intrinsic with evaluating the approach only in simulation.
Evaluating the approach in the real-world seems a natural and desirable next step.

I invite the authors to carefully integrate all the feedback received and, in particular, better highlight limitations and societal impact.

Personal comment: The references would benefit from some work: some of the references are poorly formatted; [30] and [31] cite the same paper; and finally you are not really citing 110 papers in the main text, so I think you might have forgot a \nocite in the code.

**Award:**

No

---

### Decision · Program_Chairs · 2022-09-14

Accept